# DART: DIFFERENTIABLE ADAPTIVE REGION TOKENIZER FOR VISION FOUNDATION MODELS

## ABSTRACT

The content-agnostic, fixed-grid tokenizers used by standard large-scale vision models like Vision Transformer (ViT) and Vision Mamba (Vim) represent a fundamental performance bottleneck, creating a trade-off between capturing fine-grained detail and suffering from redundant computation. To mitigate this dilemma, we introduce DART, a fully differentiable **Dynamic Adaptive Region Tokenizer**. DART employs learnable region scores and quantile-based partitioning to create content-aware patches of varying sizes, intelligently allocating a higher token density to information-rich regions. The impact of this approach is profound: it unlocks a more intelligent scaling paradigm, where a DART-equipped DeiT-Small (22M parameters) matches the performance of a DeiT-Base (86M) with nearly double the inference speed by efficiently capturing high-resolution details in key regions. Furthermore, the principle of adaptive tokenization proves its generality with clear benefits in dense prediction and spatiotemporal video tasks. We argue that by addressing the tokenizer bottleneck at its source, adaptive tokenization is a key component for building the next generation of more efficient and capable foundation models for multimodal AI, robotics, and content generation.

## 1 INTRODUCTION

A paradigm shift is underway in vision backbones for foundation models. The **uniform backbone paradigm**, a simple, non-hierarchical Vision Transformer (ViT) Dosovitskiy et al. (2021) that processes a flat sequence of image patches, has become the de facto standard. This trend is dominant across the ecosystem, from open-source Large Multimodal Models (LMMs) like the LLaVA series Liu et al. (2023) using a CLIP ViT Radford et al. (2021a), to proprietary models like Meta's Llama-3-V Yang et al. (2024) and Google's Gemini Gemini Team (2023). This convergence extends to generative models, where a major transition from CNN-based U-Nets to Transformers, catalyzed by the Diffusion Transformer (DiT) Peebles & Xie (2023), now powers flagship models like OpenAI's Sora Brooks et al. (2024) and Stability AI's Stable Diffusion 3 Esser et al. (2024). The uniform ViT's dominance is driven by its exceptional scalability and architectural simplicity.

However, this successful paradigm relies on a primitive tokenizer, creating a fundamental **representational dilemma**. The fixed-resolution patch grid simultaneously suffers from *insufficient detail* for small objects and *redundant encoding* of low-information backgrounds. A common, brute-force remedy is to increase input resolution via long-sequence fine-tuning Touvron et al. (2022). While this improves detail capture Wei et al. (2021), it drastically worsens redundancy and incurs prohibitive computational costs, creating a stark trade-off between higher fidelity and severe inefficiency.

Hierarchical architectures like Swin Transformer Liu et al. (2021) were designed to address this dilemma. By structurally merging patches to create multi-scale feature pyramids, they reduce redundancy while retaining high-resolution information in early layers. While highly successful for tasks like classification and dense prediction, this architectural philosophy has trade-offs. Its multi-scale features can mismatch the flat sequence-to-sequence structure of Large Language Models (LLMs), often requiring specialized adapters Alayrac et al. (2022). Moreover, the LMM training ecosystem's momentum around powerful, pretrained uniform ViTs (e.g., CLIP Radford et al. (2021b), SigLIP Zhai et al. (2023)) makes them a more direct and compatible choice.

This work explores an alternative path. Rather than altering the backbone, we ask: can the representational dilemma be addressed at its source, within the uniform paradigm? We propose **DART**

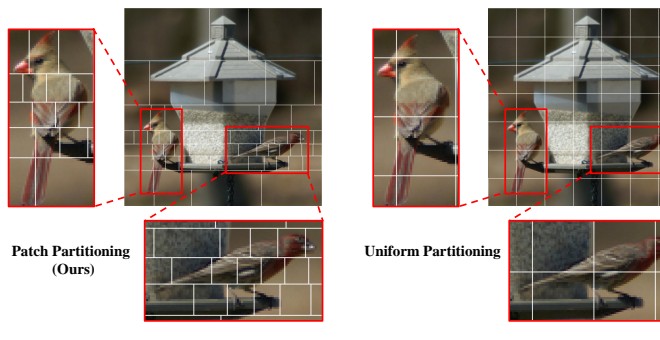

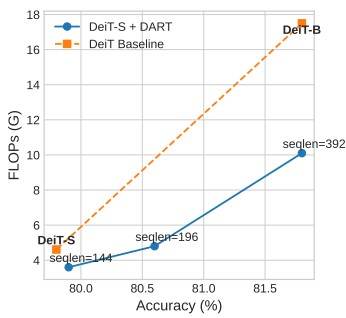

Figure 1: Comparison between our patch partitioning (left) and the uniform partitioning (right). Finer patches are allocated to the bird region, merging low-information background.

Figure 2: FLOPs vs. Accuracy trade-off curve for DeiT models.

**(Differentiable Adaptive Region Tokenizer)**, a lightweight, fully differentiable module that replaces the rigid grid with a content-aware partitioning strategy. As illustrated in Figure 1, DART allocates a higher token density to information-rich regions, capturing critical details without redundantly processing the background. It is a drop-in enhancement that preserves the ViT architecture's simplicity, scalability, and ecosystem compatibility.

Crucially, DART's impact transcends mere efficiency; it unlocks a more intelligent and efficient paradigm for performance scaling. Our results are compelling: when handling higher-resolution inputs, a DeiT-Small Touvron et al. (2021) with DART matches the baseline's performance while requiring only 46% of the computational cost. More profoundly, this enables superior "test-time scaling" over brute-force "train-time scaling." As shown in Figure 2, a DeiT-Small (22M params) Touvron et al. (2021) equipped with DART matches the 81.8% ImageNet accuracy of a much larger DeiT-Base (86M params) Touvron et al. (2021), using only a quarter of the parameters at nearly double the inference speed (**0.58x latency, 1.7x throughput**, detailed in Table 6). Similarly, a Vim-Small (29M) Zhu et al. (2024) surpasses its Vim-Base counterpart (98M) Zhu et al. (2024), demonstrating a comparable leap in efficiency. Furthermore, the principle of adaptive tokenization proves broadly applicable, delivering consistent improvements on downstream tasks ranging from dense prediction to spatiotemporal video classification. This shows that *how* a computational budget is spent is as crucial as its size, offering a more cost-effective path to high performance. Our contributions are:

- We introduce DART, a fully differentiable, content-adaptive tokenizer that significantly mitigates the core representational dilemma of the uniform ViT paradigm without altering its architecture.

- We demonstrate that DART unlocks a more intelligent scaling paradigm, enabling smaller models to match or surpass larger counterparts with substantially lower computational cost.

- We validate DART on canonical backbones like DeiT Touvron et al. (2021) and Vision Mamba Zhu et al. (2024), showing significant improvements across image and video tasks.

## 2 RELATED WORK

### 2.1 SOLVING THE TOKENIZATION BOTTLENECK: ARCHITECTURAL VS. FRONT-END SOLUTIONS

The inefficiency of the uniform tokenizer in ViT Dosovitskiy et al. (2021) has been widely recognized, leading to two distinct philosophical approaches for resolution.

**Architectural Solutions.** The first approach is architectural, exemplified by hierarchical models like PVT Wang et al. (2021) and Swin Transformer Liu et al. (2021). These models use downsample mechanisms like patch merging to create multi-scale feature pyramids, reducing spatial redundancy in deeper layers. However, this is a heavyweight solution that fundamentally alters ViT's simple, homogenous structure. This incurs the architectural costs discussed previously: complexity in multimodal fusion and a mismatch with the ecosystem of pretrained uniform ViTs. Furthermore, this

hierarchical downsampling process is **content-agnostic**; it applies a fixed, structural reduction across the entire feature map instead of dynamically allocating more resolution to critical details.

**Front-End Solutions.** In contrast, DART is a lightweight, front-end solution. Our philosophy is not to change the powerful ViT engine, but to equip it with a more intelligent intake system. By performing content-aware partitioning *before* the backbone, DART provides the ViT with an information-dense token sequence from the outset. This non-invasive approach preserves the architectural integrity and benefits of the uniform backbone paradigm, making it a "plug-and-play" enhancement rather than a fundamental redesign.

## 2.2 Dynamic Inference: Dynamic Token Reduction and Adaptive Tokenization

**Dynamic Token Reduction and Merging.** A prominent line of work focuses on alleviating the computational burden of standard ViTs by progressively reducing the token count *within* the backbone. Methods like DynamicViT Rao et al. (2021) and A-ViT Yin et al. (2022) employ learnable scoring modules to dynamically prune less informative tokens at various stages. Similarly, ToMe (Token Merging) Bolya et al. (2023) introduces a parameter-free strategy that progressively merges similar tokens via bipartite soft matching, while IA-RED$^2$ Pan et al. (2022a) utilizes a multi-level interpreter to discard redundant patches. While effective for efficiency, these methods act as a *post-hoc* remedy: they accept the initial low-quality representation from a fixed-grid tokenizer and attempt to refine it later. Furthermore, these reduction decisions, whether strictly pruning or merging, can result in variable spatial structures or necessitate masking, complicating downstream processing.

**Adaptive Tokenization.** Alternative approaches seek to address the representation bottleneck at the source by replacing the fixed grid with content-adaptive partitioning. One category of methods leverages classic computer vision algorithms to group pixels, such as Quad-trees Ronen et al. (2023), Graph Clustering Aasan et al. (2024), or Watershed algorithms Chen et al. (2025). However, these mechanisms are inherently **non-differentiable**, preventing end-to-end optimization with the backbone and often necessitating complex multi-stage training pipelines.

To achieve end-to-end differentiability, a second category of methods introduces approximation techniques. MSViT Havtorn et al. (2023) employs Gumbel-Softmax, while $\partial$HT Aasan et al. (2025) relies on a "mean-injection" trick reminiscent of **Straight-Through Estimation (STE)**. Furthermore, they inherently produce **Ragged Tensors**, which causes poor parallelization efficiency on standard hardware, often negating theoretical FLOPs reductions.

A third category, exemplified by SPFormer Mei et al. (2024), opts for a **heavyweight architectural redesign**. It employs a dual-branch structure with **iterative** cross-attention to refine superpixels. This approach fundamentally alters the ViT architecture. Moreover, its reliance on **serialized computation** for iterative refinement is unfriendly to GPU parallelism, incurring significant latency overhead.

In contrast, by leveraging a quantile-based inverse transform, DART achieves differentiability without gradient proxies and is able to output **Regular Tensors** (fixed sequence length), ensuring seamless compatibility and high throughput within standard ViT/Mamba pipelines.

## 2.3 Emerging Uniform Backbones and The Need for Adaptive Tokenization

The sequence modeling landscape is evolving, with State Space Models (SSMs) like Mamba Gu & Dao (2024) emerging as powerful alternatives to self-attention. Vision Mamba (Vim) adapts this for vision, yet as a new member of the *uniform backbone family*, it inherits ViT's primitive, fixed-patch tokenizer. This makes it a perfect case study for DART's general applicability. We posit that an intelligent front-end that efficiently converts an image into an information-rich token sequence is a fundamental need, regardless of the backbone's processing mechanism (attention, SSMs, or future ones). DART is designed as a universal solution to this challenge.

## 3 METHOD

### 3.1 OVERALL PIPELINE

Our dynamic, content-aware tokenizer replaces the conventional fixed-grid patcher via a three-stage process. First, a lightweight scoring network predicts a map of information density. Second, a differentiable partitioning module uses this map to compute adaptive patch boundaries. Third, the resulting non-uniform patches are resampled to a fixed size and projected into tokens. For a detailed visualization of the architecture and data flow, please refer to Figure 12 in the Appendix. This partitioning module has two variants: a grid-preserving method and a more advanced topology-breaking one.

**Differentiable Quantile Computation.** Our differentiable quantile method is based on inverting the Cumulative Distribution Function (CDF) of a 1D discrete probability distribution $S = \{S_i\}_{i=0}^{\text{seqlen}-1}$. We model this distribution as a histogram (Figure 3a), which yields a piecewise-linear, monotonically increasing CDF. The quantile boundaries are found by inverting this function. For each target cumulative probability $q_k = k/N$, we locate the linear segment of the CDF containing it and then compute the precise corresponding boundary point. Since this inversion process is differentiable almost everywhere, the resulting partition boundaries are differentiable with respect to the input distribution $S$, enabling end-to-end training (a detailed mathematical formalism is provided in Appendix E.2).

### 3.2 SCORE PREDICTION NETWORK

Given an input image $X \in \mathbb{R}^{H \times W \times 3}$, a lightweight, CNN extracts a feature map $F \in \mathbb{R}^{H' \times W' \times C}$, from which a trainable MLP (scoring head) predicts a single-channel score map $\{s_{i,j}\}$. To produce a stable 2D probability distribution $\{\tilde{s}_{i,j}\}$, these raw scores undergo sequential normalization: a sigmoid constrains values to $[0, 1]$, and a per-sample normalization ensures they sum to 1. This final distribution quantifies each location's relative importance.

### 3.3 DART-GRID: GRID-PRESERVING PARTITIONING

The most intuitive adaptive partitioning approach creates a non-uniform grid that preserves the original patch topology. We call this method **DART-Grid**. It independently partitions the horizontal and vertical axes based on their marginal probability distributions:

1. The marginal distributions for the y-axis ($P_Y$) and x-axis ($P_X$) are computed by summing the 2D probability distribution $\{\tilde{s}_{i,j}\}$ along each axis.

2. Our 1D differentiable quantile algorithm is applied to $P_Y$ to find $N_h - 1$ horizontal boundaries, defining $N_h$ rows of varying heights.

3. The same algorithm is similarly applied to $P_X$ to find $N_w - 1$ vertical boundaries, defining $N_w$ columns of varying widths.

The result is a content-aware $N_h \times N_w$ grid where each patch's area is inversely proportional to its region's information density, while the overall grid structure remains intact, as shown in Figure 3b.

### 3.4 DART-FLOW: TOPOLOGY-BREAKING PARTITIONING WITH TOKEN FLOW

While DART-Grid adapts patch sizes, it constrains the token budget within a rigid grid structure. For a more powerful, global reallocation of resources, we propose **DART-Flow**, an advanced version that breaks this topology. As our primary contribution, it is referred to simply as **DART** throughout this paper and operates in two sequential stages:

1. **Adaptive Row Partitioning:** First, we partition the image horizontally into $N_h$ rows of varying heights, identical to the y-axis partitioning step in DART-Grid. This initial step allocates vertical space based on horizontal information density.

2. **Global Token Allocation via Virtual Flattening:** Next, we conceptually concatenate these $N_h$ adaptive rows into a single, long 1D sequence. The probability distribution over

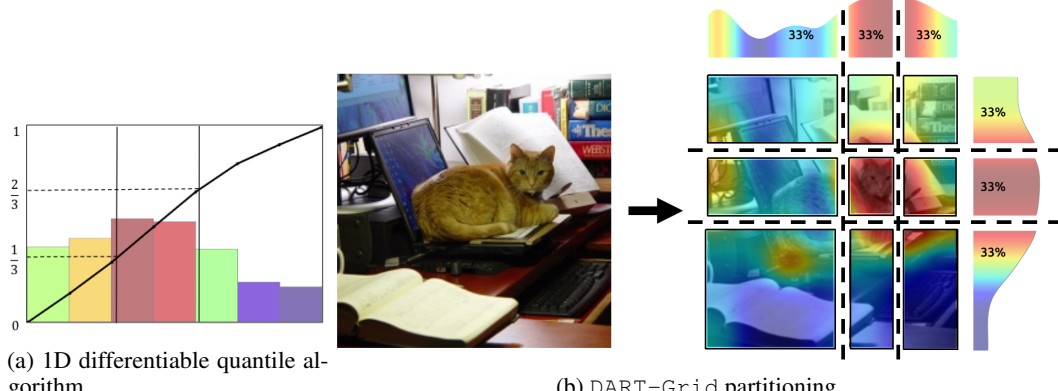

(a) 1D differentiable quantile algorithm.

(b) `DART-Grid` partitioning.

Figure 3: Illustration of our core differentiable partitioning mechanism. **(a)** Boundaries for a 1D distribution are found by inverting its piecewise-linear CDF at uniform quantiles (e.g., 1/3, 2/3). **(b)** The `DART-Grid` is formed by independently applying this 1D algorithm to the horizontal and vertical marginal distributions of the 2D score map, where the score map produced by the lightweight network is overlaid on the image.

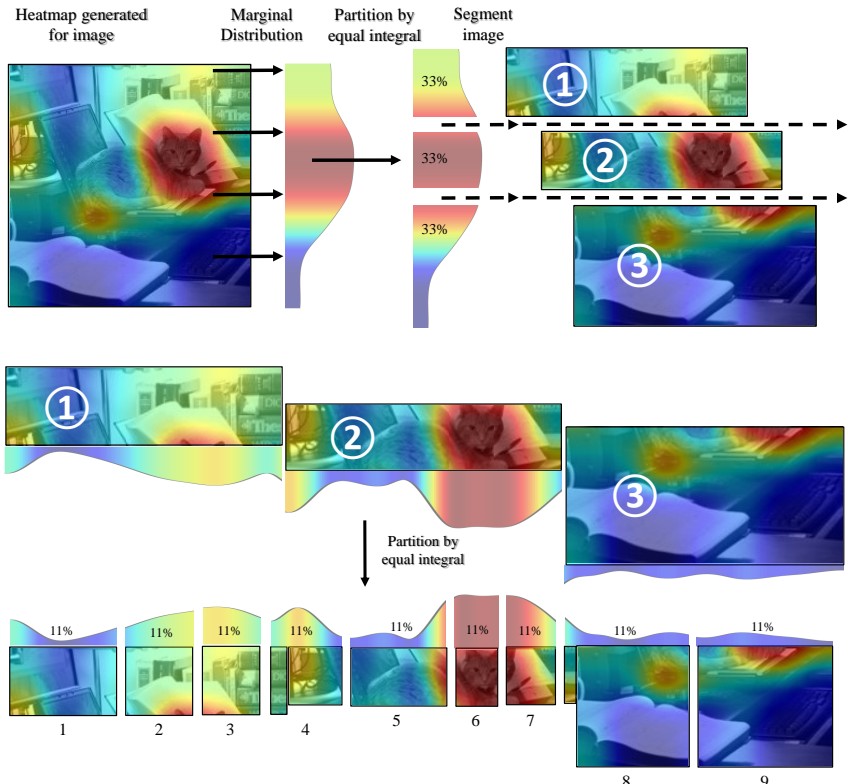

Figure 4: The **DART-Flow** process. The partitioning algorithm is applied sequentially: first to create adaptive horizontal rows, and then to globally allocate tokens across all rows.

> this flattened sequence is derived from the original 2D score map. We then apply our 1D differentiable quantile algorithm *once* to this long sequence to find all $N_{total} - 1$ boundaries for the final patches.

This topology-breaking design is DART's key innovation. It allows the total token budget ($N_{total}$) to be distributed globally, enabling a **"flow"** of tokens away from low-information rows and concentrating them in high-information rows, far beyond the constraints of a simple grid, as shown in Figure 4.

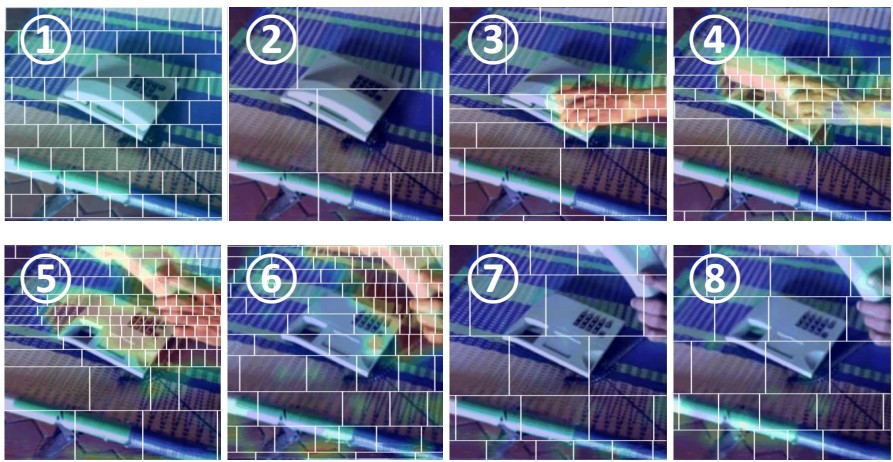

Figure 5: Patch partitioning on an example from the SSv2 dataset Goyal et al. (2017).

### 3.5 APPLICATION TO VIDEO

Our partitioning approach extends naturally to video by vertically concatenating all frames into a composite image and applying our dynamic partitioning. This enables the uneven distribution of a fixed token budget across space and time, concentrating resources on key frames or critical regions. To capture temporal importance, the scoring network processes the feature difference between consecutive frames, learning to focus on motion. This design has an ideal effect: a stationary object is encoded intensively only when it first appears. As it remains unchanged in subsequent frames, it receives few computational resources, achieving efficient temporal redundancy compression, as shown in Figure 5.

### 3.6 DIFFERENTIABLE RESAMPLING AND POSITIONAL TRANSFORMATION

Once patch boundaries are determined, a differentiable sampling module converts the non-uniform regions into fixed-size tokens. For image content, a regular $16 \times 16$ target grid is defined for each output token. DART's dynamic boundaries define an affine transformation mapping this grid to locations on the input image, from which a standardized $16 \times 16$ patch is sampled via bilinear interpolation. These patches, created through a fully differentiable process, are then linearly projected into tokens. To maintain spatial awareness, positional embeddings (PE) are transformed similarly. The PE is treated as a learnable, low-resolution map ($H_{pe} \times W_{pe} \times D$), and each token's PE is derived by sampling from this map at the token's center coordinate using bilinear interpolation. This critical step preserves spatial relationships by informing the model of each token's origin. The entire system is end-to-end differentiable and trained with a standard cross-entropy loss.

## 4 EXPERIMENTS

### 4.1 EXPERIMENTAL SETUP

Our main image classification experiments are conducted on the ImageNet-1K ILSVRC-2012 dataset Deng et al. (2009), using Top-1 accuracy as the primary metric. The evaluation focuses on the uniform backbone family, with DeiT (Transformer-based) and Vision Mamba (Vim) (SSM-based) as our core baselines. For a fair comparison, all training hyperparameters strictly follow the official configurations of the baseline models, with details provided in the Appendix. All experiments were performed on a single machine with eight NVIDIA A100 GPUs.

### 4.2 THE CORE THESIS: UNLOCKING AN INTELLIGENT SCALING PARADIGM

A primary method for enhancing Vision Transformer performance is long-sequence fine-tuning, which increases input resolution. However, this brute-force approach of uniformly increasing token density leads to a quadratic explosion in computational cost. We posit that DART's content-aware

Table 1: Unprecedented efficiency on high-resolution inputs. † denotes long-sequence fine-tuning.

| Backbone | Tokenizer | Params | Patches | FLOPs | Top-1 (%) |
|----------|-----------|--------|---------|-------|-----------|
| DeiT-S† | Baseline | 22M | 576 | 15.5G | 81.6 |
| DeiT-S† | **DART** | 24M | **288** | **7.2G** | **81.5** |
| VideoMamba-Ti† | Baseline | 7M | 1296 | 7.11G | 79.6 |
| VideoMamba-Ti† | **DART** | 8M | **392** | **2.24G** | **79.7** |

Table 2: A superior path to top-tier performance. † denotes long-sequence fine-tuning.

| Backbone | Params | Patches | FLOPs | Top-1 (%) |
|----------|--------|---------|-------|-----------|
| *DeiT Family* | | | | |
| DeiT-B (Target) | 86M | 196 | 17.5G | 81.8 |
| DeiT-S† | 22M | 576 | 15.5G | 81.6 |
| DeiT-S† + **DART** | 24M | **392** | **10.1G** | **81.8** |
| *Vim Family* | | | | |
| Vim-B (Target) | 98M | 196 | 19.9G | 81.9 |
| Vim-S† | 26M | 784 | 19.6G | 81.6 |
| Vim-S† + **DART** | 29M | **392** | **10.9G** | **82.2** |

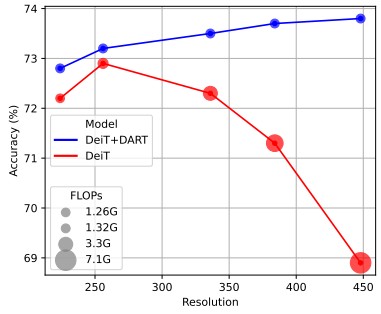

Figure 6: Ablation on input resolution.

token allocation enables a more intelligent form of test-time scaling. This new paradigm offers two key benefits: unprecedented computational efficiency on high-resolution inputs, and a superior path to top-tier performance that avoids training larger models.

First, we demonstrate DART's value in high-resolution fine-tuning. As shown in Table 1 and Figure 2, a DART-equipped DeiT-S matches a conventional baseline's performance (81.5%) with only 46% of the computational cost (7.2 vs. 15.5 GFLOPs). This efficiency principle is broadly applicable and sometimes even more dramatic; on VideoMamba-Ti Li et al. (2024), for example, DART delivered similar accuracy using just 31% of the computational cost.

DART also enables a fundamentally better scaling path, as shown in Table 2. While standard high-resolution fine-tuning is costly and fails to elevate small models (DeiT-S, Vim-S) to match their larger Base counterparts, DART-equipped models succeed. A DART-powered DeiT-S matches the accuracy of DeiT-Base, and a Vim-S with DART surpasses its Vim-Base counterpart. This suggests that using DART to efficiently process high-resolution data with a smaller model is a more effective strategy than simply training a larger one.

This flexibility is enabled by DART's facilitation of true *test-time scaling*. The presented results are from a single model checkpoint fine-tuned on longer sequences. Thanks to DART's dynamic partitioning, this single model is exceptionally flexible, able to seamlessly process various sequence lengths at inference time, including non-square numbers like 288 or 392. Crucially, when this fine-tuned model is fed the original 196-token input, its performance drops by only 0.2% from its pre-fine-tuning level. This "train once, use flexibly" capability allows users to select their desired accuracy-compute trade-off at test time, embodying a truly dynamic scaling solution.

This demonstrates a more cost-effective and sustainable path to high performance. Consequently, our investigation focuses on this intelligent scaling strategy rather than pursuing costly standard-configuration experiments on the Base-scale models, as our approach already provides a more efficient means to achieve their performance levels.

### 4.3 FOUNDATIONAL VALUE: A UNIVERSAL ENHANCEMENT

Having established DART's ability to unlock a new scaling paradigm, we now verify its foundational value as a universal enhancement in standard, fixed-budget scenarios. We evaluate DART as a simple drop-in replacement for the standard tokenizer, keeping the total token count fixed at 196. While this enhancement introduces a minimal overhead of approximately 5% (see Appendix B.2 for

Table 4: Performance gains as a drop-in module.

| Backbone | Tokenizer | Params | FLOPs | Top-1 (%) |
|---|---|---|---|---|
| DeiT-Ti | Baseline | 6M | 1.26G | 72.2 |
| DeiT-Ti | **DART** | 7M | 1.32G | 73.8 **(+1.6)** |
| DeiT-S | Baseline | 22M | 4.61G | 79.8 |
| DeiT-S | **DART** | 24M | 4.84G | 80.6 **(+0.8)** |
| Vim-Ti | Baseline | 7M | 1.60G | 76.1 |
| Vim-Ti | **DART** | 8M | 1.68G | 77.2 **(+1.1)** |
| Vim-S | Baseline | 26M | 5.30G | 80.5 |
| Vim-S | **DART** | 29M | 5.55G | 81.5 **(+1.0)** |
| VideoMamba-Ti | Baseline | 7M | 1.08G | 76.9 |
| VideoMamba-Ti | **DART** | 8M | 1.15G | 78.2 **(+1.3)** |

Table 3: Semantic segmentation on the ADE20k validation set.

| Backbone | mIoU (%) |
|---|---|
| Swin-T (Baseline) | 44.5 |
| **+ DART-Grid** | 45.0 **(+0.5)** |

details), Table 4 shows it yields consistent performance improvements across all tested Transformer and SSM-based backbones. The improvement is particularly notable on DeiT-Ti, which sees a +1.6% increase in Top-1 accuracy. One potential reason is that improving upon a lower-performance baseline can result in a larger gain in the absolute accuracy metric as we observe that DeiT-Ti's baseline performance (72.2%) is considerably lower than other similarly-scaled models such as Vim-Ti (76.1%) and VideoMamba-Ti (76.9%).

## 4.4 GENERALIZING THE PRINCIPLE AND POSITIONING AGAINST ALTERNATIVES

Having established DART's value in image classification, this section explores its universality. We extend our evaluation to dense pixel-level prediction and spatiotemporal video classification, followed by ablation studies that provide deeper insights into the source of its gains.

### 4.4.1 CASE STUDY: GENERALIZATION TO DENSE PREDICTION

To validate DART on dense prediction, we integrate it into a UPerNet Xiao et al. (2018) with a Swin-T backbone on the ADE20k dataset Zhou et al. (2017). Swin Transformer presents a unique case, as its design already addresses the detail-redundancy trade-off by merging small early-layer tokens into larger ones in deeper layers. However, this merging process is **structural and content-agnostic**.

DART's content-aware approach has a similar goal, making their performance gains **partially non-orthogonal**, which suggests a more modest improvement than on a standard ViT. Since Swin Transformer's mechanisms (e.g., windowed attention and patch merging) require a regular grid, our topology-breaking DART-Flow method is incompatible. We therefore used our grid-preserving DART-Grid variant. As shown in Table 3, despite the non-orthogonal gains, adding `DART-Grid` still yields a **+0.5 mIoU** improvement. This confirms that content-aware tokenization provides tangible benefits even on a strong baseline with a structural multi-scale strategy.

### 4.4.2 CASE STUDY: EXTENSION TO THE SPATIOTEMPORAL DOMAIN

DART's effectiveness in the spatiotemporal domain is highlighted by the results in Table 5. On the motion-reliant Something-Something-V2 (SSv2) dataset, our mechanism improves accuracy by +0.5% while cutting GFLOPs by 41% by allocating tokens to the most informative moments. On the more scene-centric Kinetics-400 Carreira & Zisserman (2017), DART still provides a robust +0.4% accuracy gain. Notably, these gains are achieved with a feature extractor whose weights are frozen after pretraining on ImageNet, demonstrating that its learned features for identifying salient regions are general enough to transfer effectively to new domains.

### 4.5 COMPARISON WITH DYNAMIC INFERENCE METHODS

While motivated by the shared goal of improving efficiency, DART's approach of optimizing tokens at their source is fundamentally different from methods that adapt the sequence *internally* by pruning or merging tokens. Thanks to its flexible sequence length, DART allows for direct, fair comparisons at similar computational budgets. As detailed in Table 7, DART consistently outperforms prior methods.

Table 5: Video classification results on SSv2 and Kinetics-400.

| Dataset | Method | Patches | Top-1 (%) |
|---|---|---|---|
| SSv2 | VideoMamba-Ti | 1568 | 63.2 |
| SSv2 | **+ DART** | 784 | **63.7 (+0.5)** |
| Kinetics-400 | VideoMamba-Ti | 1568 | 76.9 |
| Kinetics-400 | **+ DART** | 1568 | **77.3 (+0.4)** |

Table 6: Inference speed comparison on a 3090 GPU (batch size 512).

| Model | Patches | Latency | Img/s |
|---|---|---|---|
| DeiT-B | 196 | 1951 ms | 262 |
| DeiT-S | 576 | 1575 ms | 325 |
| +DART | 392 | 1142 ms | 448 |
| +DART | 288 | 814 ms | 629 |

Table 7: Comparison with dynamic inference methods Yin et al. (2022); Havtorn et al. (2023); Rao et al. (2021); Pan et al. (2022a).

| Method | Backbone | Patches | FLOPs | Top-1 (%) |
|---|---|---|---|---|
| A-ViT | DeiT-Ti | dynamic | 0.8G | 71.0 |
| **DART (Ours)** | DeiT-Ti | 121 | 0.8G | **71.8** |
| MSViT | DeiT-S | dynamic | 3.3G | 78.8 |
| A-ViT | DeiT-S | dynamic | 3.6G | 78.6 |
| **DART (Ours)** | DeiT-S | 144 | 3.6G | **79.9** |
| DynamicViT | DeiT-S | dynamic | 7.0G | 80.3 |
| **DART (Ours)** | DeiT-S | 288 | 7.2G | **81.5** |
| DynamicViT | DeiT-B | dynamic | 11.2G | 81.3 |
| $IA-RED^2$ | DeiT-B | dynamic | 11.8G | 80.3 |
| **DART (Ours)** | DeiT-S | 392 | 10.1G | **81.8** |

Table 8: Ablation on partitioning strategies using DeiT-Ti as the backbone.

| Method | Top-1 (%) |
|---|---|
| Deit-Ti | 72.2 |
| + DART-Grid | 73.1 |
| **+ DART-Flow** | **73.8** |

At similar GFLOPs, DART surpasses A-ViT and DynamicViT. More notably, when compared against methods for the DeiT-Base backbone, our approach using the smaller DeiT-S backbone achieves superior performance at a lower computational cost.

## 4.6 ANALYSIS AND ABLATION STUDIES

**Ablation on Partitioning Strategy.** We compare our main topology-breaking DART-Flow with the simpler, grid-preserving DART-Grid. While DART-Grid improves over the baseline, DART-Flow is significantly better (Table 8), confirming the superiority of its design for global token reallocation. As a crucial sanity check, we verified experimentally that a uniform score map degenerates our mechanism to the standard ViT tokenizer, matching its baseline performance and confirming that all gains stem from the learned partitioning itself.

**Visualization of the Learning Process.** Figure 7 visualizes the evolution of DART's learned score map. This smooth, gradual refinement is a direct consequence of our **fully differentiable design**, which allows the model to fluidly learn **where** to allocate its computational budget via end-to-end optimization. The visualization clearly shows the model learning to concentrate tokens on the foreground object over time, validating our approach.

**Ablation on Input Resolution.** To verify that DART's gains stem from its fine-grained partitioning, we ablated the input resolution while keeping the token count fixed. Our hypothesis is that resampling

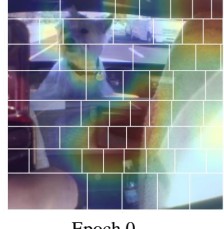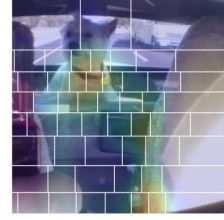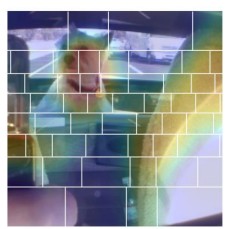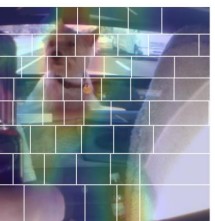

| Epoch 0 | Epoch 40 | Epoch 70 | Epoch 300 |

Figure 7: Visualization of the learned score map and patch boundaries at different training epochs, showing the model progressively learning to focus on the salient object.

error is primarily confined to the small, dense patches in critical regions. Large background patches have ample source pixels and thus minimal error even at low input resolutions. The real bottleneck is the fidelity of critical patches when they are upsampled to the fixed $16 \times 16$ grid. The results in Figure 6 confirm this: performance increases as resolution scales from 224 to 448 because the higher pixel density reduces information loss for these bottleneck patches. Performance saturates beyond $448 \times 448$, as the source resolution is now high enough to faithfully represent even the smallest patches, justifying our use of this resolution for main experiments.

## 5 CONCLUSION

We introduced DART, a content-aware tokenizer that addresses key inefficiencies of rigid tokenization in uniform backbones like ViT and Mamba. More than an incremental improvement, DART unlocks an intelligent scaling paradigm, empowering smaller models to match or surpass counterparts four times their size at a fraction of the computational cost. A core principle emerges from this work: investing a small computational overhead to scout the input allows for a much more efficient allocation of the main processing budget. By offering a more cost-effective path to high performance, adaptive tokenization is poised to be a key component for the next generation of foundation models in multimodal AI, robotics, and content generation. We discuss promising directions for future work and the broader implications of our approach in Appendix A.

## REPRODUCIBILITY STATEMENT

To ensure the reproducibility of our research, we have made a comprehensive effort to provide all necessary components. Our approach and resources are organized as follows:

- **Source Code:** The complete source code for our DART module is provided as a ZIP file in the supplementary materials. The code includes a detailed README file with instructions for setting up the environment and running the experiments. We will release the code publicly upon acceptance.

- **Methodology and Implementation:** The core methodology of DART is described in detail in Section 3. To further clarify the implementation, we provide pseudocode for our main topology-breaking tokenizer (`DART-Flow`) in the Appendix (Figure 13). Specific architectural details for the scoring networks used with different backbones are listed in Appendix Table 13.

- **Training Configurations:** Our main experimental setup is outlined in Section 4. We strictly followed the official training configurations of the baseline models to ensure fair comparisons. Any modifications, specifically the hyperparameters for our long-sequence fine-tuning experiments, are explicitly documented in Appendix C.2.

- **Datasets and Baselines:** All experiments were conducted on standard, publicly available datasets, including ImageNet-1K, ADE20k, Something-Something-V2, and Kinetics-400. We utilized well-established open-source implementations for all baseline models (e.g., DeiT, Vision Mamba), and followed their standard data preprocessing pipelines.

- **Computational Environment:** Our primary experiments were conducted on a server with eight NVIDIA A100 GPUs. Details on the inference benchmarks, including latency and throughput measurements, which were performed on a single NVIDIA RTX 3090 GPU, are provided in Table 6 and Appendix Table 10.

We believe these resources provide a clear and complete roadmap for replicating our findings.

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

## A    DISCUSSION AND FUTURE WORK

First, our work establishes DART as a foundational component for efficient visual representation. A significant and promising future direction is to integrate DART as the vision front-end for large-scale systems such as Large Multimodal Models, Embodied AI agents with Vision-Language-Action models, and generative models. In these systems, where visual inputs are often complex and continuous, DART's ability to intelligently allocate computational resources is not just beneficial but potentially critical for achieving real-time performance and better grounding. This paper provides the foundational validation, paving the way for exploring these advanced applications.

Second, our current approach employs a frozen, pretrained scoring network for general applicability. This design choice opens a clear path for domain-specific adaptation. For specialized applications such as medical or satellite imagery, where the definition of "information-rich" regions is domain-specific, fine-tuning the scoring network on task-relevant data presents a compelling opportunity to further boost performance. This highlights the adaptability of the DART framework to specialized downstream tasks.

Third, the current framework masterfully optimizes the token budget within each sample. This naturally opens the door to a more advanced paradigm: inter-sample dynamic allocation, where the total token budget could vary based on sample complexity. Crucially, DART's proven ability to handle arbitrary, non-square sequence lengths provides the essential technical foundation for this next step. Future models could leverage this capability to allocate more computation to "hard" examples and less to "easy" ones, further pushing the boundaries of model efficiency.

Fourth, while our primary method, `DART-Flow`, is tailored for the uniform backbone paradigm, the core principle of content-aware tokenization remains broadly applicable. We demonstrate this with our compatible `DART-Grid` variant, which successfully enhances hierarchical models like Swin Transformer, as shown in our dense prediction experiments. This not only confirms the versatility of our approach but also points to a promising research avenue: co-designing novel adaptive tokenizers that are even more deeply integrated with the architectural priors of hierarchical models.

Fifth, we consider the potential applicability of DART to low-level vision and generative tasks, specifically Super-Resolution (SR). Standard Transformer-based SR models often face computational bottlenecks when processing ultra-high-resolution inputs (e.g., 4K/8K). However, the requirement for detail refinement is typically spatially sparse, with perceptual quality hinging largely on high-frequency structures rather than smooth background regions. DART offers a mechanism to decouple input resolution from the computational budget. By adapting the scoring network to identify high-frequency artifacts, DART provides a feasible pathway to process high-resolution inputs using a fixed, tractable token count, thereby extending the efficiency benefits of adaptive tokenization to generative domains.

Finally, this paper's primary contribution is the validation of a more intelligent and compute-efficient scaling *principle*. Our extensive experiments on widely-used model sizes robustly support this principle. We hypothesize that these significant efficiency gains will persist at extreme scales (e.g., on ViT-L/H), and empirically verifying this remains a compelling direction for future work with access to large-scale computational resources. This positions DART not just as a method, but as a generalizable strategy for building more scalable and efficient foundation models.

## B    ADDITIONAL EXPERIMENTS AND ANALYSIS

### B.1    IMPACT OF THE SCORING NETWORK.

DART is robust to the choice of scoring network, though a stronger scorer yields better performance (Table 9). For instance, on DeiT-Ti, using EfficientNet-B0 instead of MobileNetV3-S boosts the accuracy gain from +1.6% to +2.9%. This presents a flexible trade-off between the tokenizer's overhead and the final model's accuracy.

Table 9: Impact of the scoring network architecture on DeiT-Ti.

| Scoring Network | FLOPs (G) | Top-1 (%) |
|---|---|---|
| w/o (Baseline) | 1.26 | 72.2 |
| MobileNetV3-S Howard et al. (2019) | 1.32 | 73.8 **(+1.6)** |
| MnasNet Tan et al. (2019) | 1.37 | 74.0 **(+1.8)** |
| SqueezeNet Iandola et al. (2017) | 1.54 | 74.3 **(+2.1)** |
| EfficientNet-B0 Tan & Le (2019) | 2.41 | 75.1 **(+2.9)** |

## B.2 OVERHEAD ANALYSIS OF THE DART MODULE

To isolate and quantify the computational overhead introduced by the DART module itself, we compared the inference performance of a standard DeiT-S with a DART-equipped model under an **identical configuration** (i.e., 196 tokens). Tests were conducted on an NVIDIA RTX 3090 GPU with a batch size of 128. As shown in Table 10, DART's tokenizer introduces a minimal latency overhead of approximately **6ms**, resulting in a throughput decrease of about **5%**. This minor overhead confirms DART's efficiency as a lightweight front-end module.

Table 10: DART module overhead on an NVIDIA RTX 3090 (196 tokens, batch size 128).

| Model | Throughput (img/s) | Latency (ms) |
|---|---|---|
| DeiT-S (Baseline) | 1110.5 | 115.3 |
| DeiT-S + DART | 1053.2 **(-5.2%)** | 121.5 **(+6.2ms)** |

## B.3 GENERALIZATION TO FINE-GRAINED CLASSIFICATION

To validate DART's ability to capture subtle discriminative details, we evaluated it on the fine-grained CUB-200-2011 dataset using linear probing. As shown in Table 11, DART-equipped DeiT-S achieves **76.2%** accuracy, surpassing the baseline (75.4%) by **+0.8%**. This confirms that DART effectively concentrates the token budget on critical local features (e.g., beaks, feathers) rather than the background.

Table 11: Linear probing accuracy on CUB-200-2011.

| Method | Backbone | Top-1 (%) | Gain |
|---|---|---|---|
| Baseline | DeiT-S | 75.4 | - |
| **DART (Ours)** | DeiT-S | **76.2** | **+0.8** |

## B.4 ABLATION ON TRAINING STRATEGY

To validate our design choice of using a frozen feature extractor with a trainable scoring head, we compared it against a full end-to-end fine-tuning strategy where both the extractor and head are updated. As shown in Table 12, full fine-tuning yields only a marginal gain (+0.1%) over our standard strategy. This confirms that the "frozen extractor + trainable head" design is highly efficient and sufficient for learning effective attention policies.

Table 12: Training strategy ablation on DeiT-Ti.

| Strategy | Top-1 (%) |
|---|---|
| Baseline | 72.2 |
| Standard (Frozen Extractor + Trainable Head) | 73.8 (+1.6) |
| Full E2E (Trainable Extractor + Trainable Head) | **73.9 (+1.7)** |

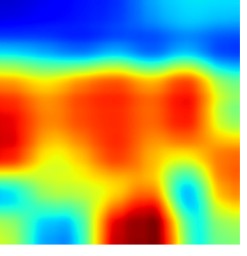 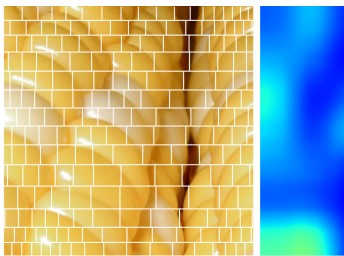

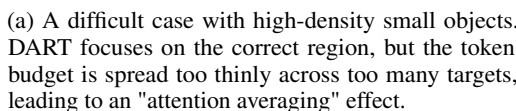

(a) A difficult case with high-density small objects. DART focuses on the correct region, but the token budget is spread too thinly across too many targets, leading to an "attention averaging" effect.

(b) A simple case where the subject fills the entire frame. DART exhibits "adaptive degeneration," reverting to a near-uniform grid, as no specific region warrants higher importance.

Figure 8: Qualitative comparison of DART's behavior on two distinct types of challenging inputs. These cases highlight the model's adaptive nature, showing how it intelligently shifts between highly focused and uniform partitioning strategies based on image content.

## C  EXPERIMENTAL SETUP

### C.1  ARCHITECTURE DETAILS

Our **DART** module requires a small pretrained feature extractor to obtain general image features for region score prediction. The specific feature extractors used by each model in our experiments are summarized in Table 13.

Table 13: Feature extractor configurations for various backbone models.

| Model | Feature Extractor |
|---|---|
| DeiT-Ti | MobileNetV3 Small[:11] |
| DeiT-S | MobileNetV3 Large[:15] |
| Vim-Ti | MobileNetV3 Small[:13] |
| Vim-S | MobileNetV3 Large[:17] |
| VideoMamba-Ti | MobileNetV3 Small[:13] |

### C.2  TRAINING CONFIGURATIONS

To ensure a fair and direct comparison, our experimental setup adheres closely to the established training protocols of the baseline models. Unless otherwise specified, all models were trained using the official, publicly available configurations provided by the original authors of DeiT, Vim, VideoMamba, and Swin Transformer.

The only exception is the long-sequence fine-tuning experiments conducted on DeiT-S, as detailed in Section 4.2. For these specific experiments, we used the following fine-tuning hyperparameters, which can be summarized with the command-line arguments: '–epochs 30 –weight-decay 1e-8 –lr 5e-6 –warmup-epochs 0 –sched step –decay-rate 1'.

## D  QUALITATIVE ANALYSIS AND VISUAL COMPARISON

### D.1  QUALITATIVE ANALYSIS OF CHALLENGING CASES

To further probe the behavior of DART, we present a qualitative analysis of two challenging cases that reveal both its strengths and inherent limitations. These examples demonstrate how the model dynamically adapts its tokenization strategy in response to vastly different image characteristics.

**Case 1: The "Attention Averaging" Bottleneck in High-Density Scenes.**  Figure 8a presents a difficult sample where both the baseline and DART models failed to classify correctly at the standard

196 token length. The challenge lies in the multitude of small, densely packed shoe objects. However, it is crucial to distinguish the nature of this failure. While the baseline model wastes a significant portion of its tokens on irrelevant background, DART successfully concentrates its entire token budget on the correct region of interest. This represents a far more effective, albeit still insufficient, use of resources. The core issue remains that its strategy of attending to all targets simultaneously becomes suboptimal when the number of targets is excessively large, leading to an "attention averaging" effect that prevents any single object from being addressed in high fidelity.

This case highlights a key difference from human visual attention, which can inspire future work. A human would likely fixate on one or two representative shoes to identify them, and then infer the similarity of the surrounding objects with lower attention. This "sampling" strategy offers a potential shortcut for classification. Interestingly, this limitation is primarily contextualized within such classification tasks; in dense prediction, which inherently demands exhaustive analysis of all objects and typically operates at higher resolutions, DART's tendency to cover all targets is not a drawback but an aligned strategy. Ultimately, this difficult sample does not reveal a new flaw introduced by our method, but rather defines the boundary of its capabilities. It illustrates an inherent challenge that DART addresses more effectively than its baseline counterpart, pointing towards even more sophisticated, human-like attention mechanisms as a future possibility.

**Case 2: "Adaptive Degeneration" on Global-Feature Samples.** In stark contrast, Figure 8b illustrates a scenario where the image is characterized by a global texture, lacking a distinct foreground or background. In this situation, DART exhibits a robust behavior we term "adaptive degeneration." The scoring network produces a relatively uniform score map, correctly assessing that no specific sub-region is more informative than another. Consequently, the partitioning algorithm naturally yields a near-uniform grid, effectively degenerating to the standard ViT tokenizer's strategy.

## D.2 QUALITATIVE ANALYSIS OF PARTITIONING

To provide a more intuitive understanding of DART's behavior, we present additional partitioning visualizations on images from the ImageNet-1K validation set in Figures 9 and 10. These examples further demonstrate that DART successfully learns to allocate a denser token budget to information-rich areas, such as object contours and complex textures, while merging low-information background regions into larger patches.

## D.3 PARTITIONING STRATEGY COMPARISON

In our main paper, we introduced two partitioning strategies: the grid-preserving `DART-Grid` and the more advanced, topology-breaking `DART-Flow`. Figure 11 provides a clear visual comparison. The `DART-Grid` method (middle) adapts patch sizes but is constrained to a fixed row-column structure. This approach is effective for objects with relatively convex shapes, such as the hartebeest (top row), where salient features are somewhat vertically aligned. However, it struggles with more complex geometries like the snake (bottom row). The reason is that row heights and column widths are adjusted globally; the method cannot efficiently focus tokens when the regions of interest in each row do not fall within the same set of columns.

In contrast, our main `DART-Flow` method (left) significantly mitigates this issue. By allowing the token budget to "flow" across row boundaries, it can concentrate tokens in the most salient regions without being constrained by the grid topology, effectively tracing the object's shape.

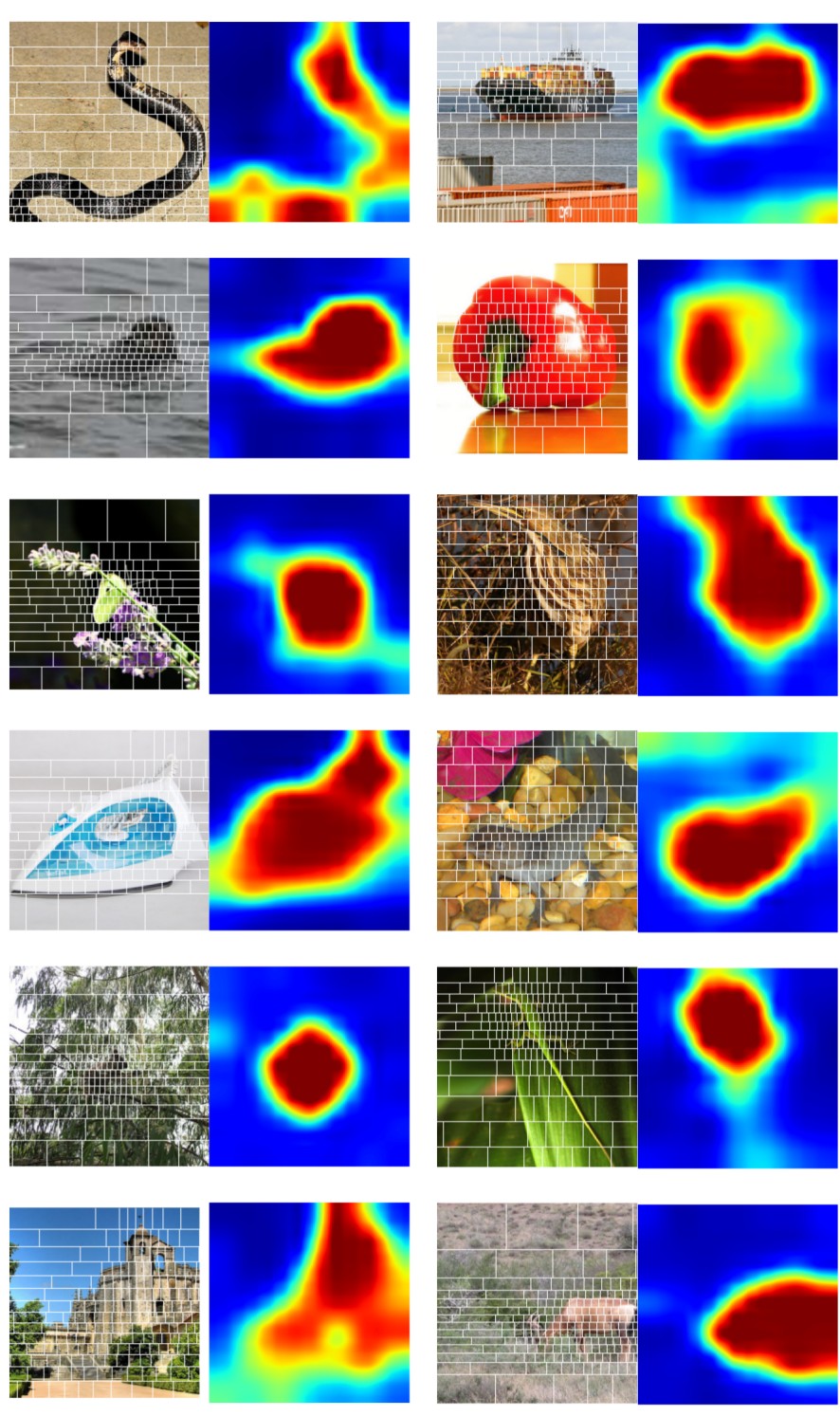

Figure 9: Partition examples produced by our DART model (Part 1).

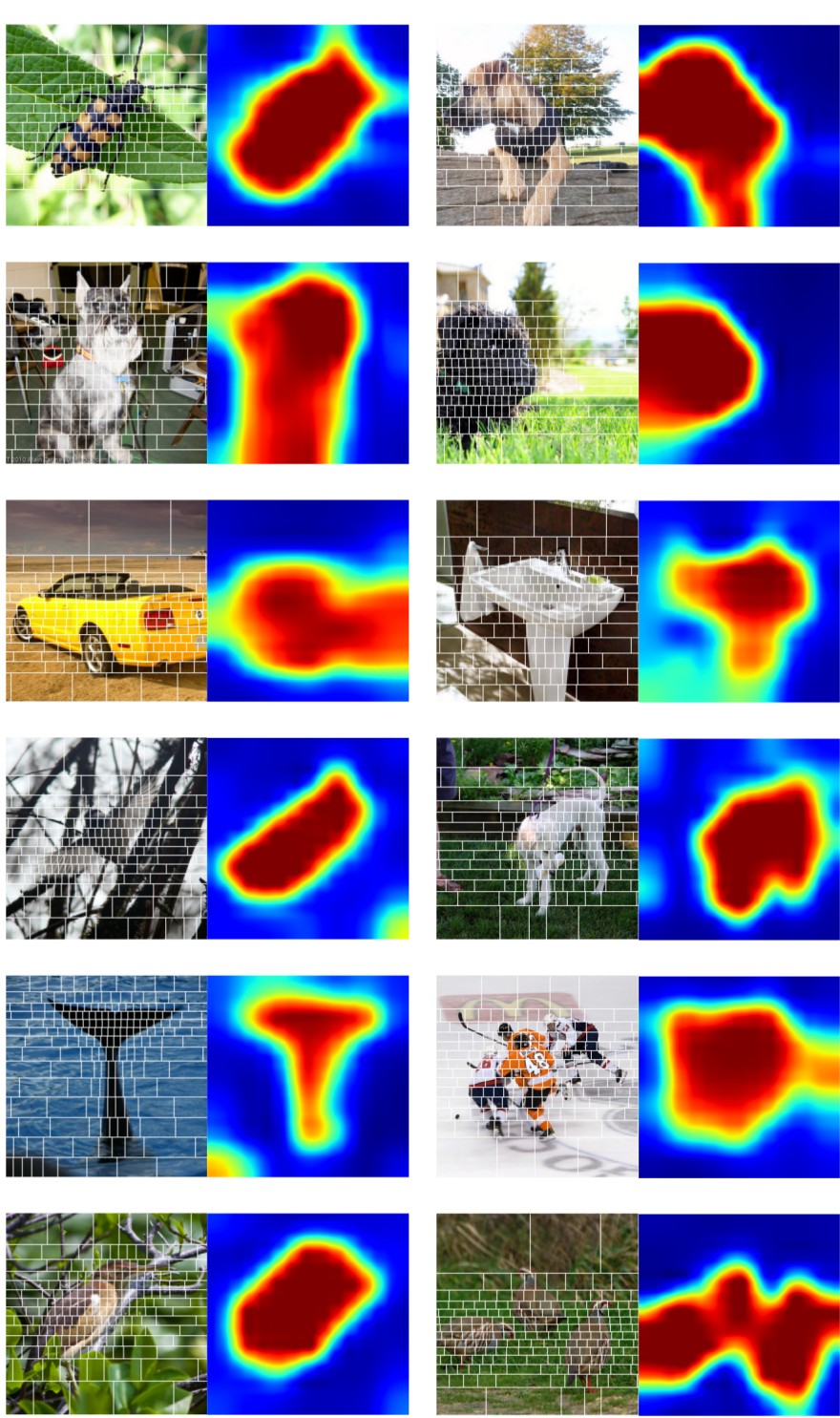

Figure 10: Partition examples produced by our DART model (Part 2).

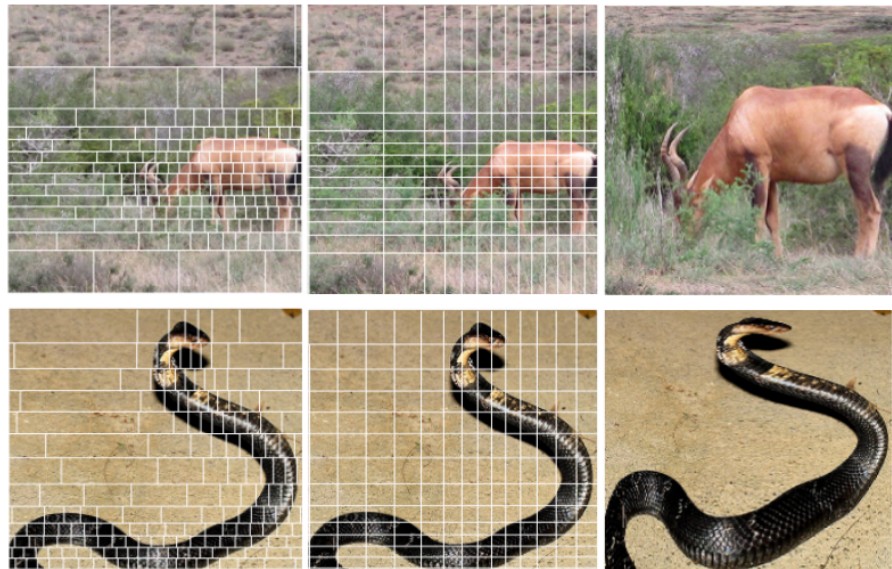

Figure 11: A visual comparison of partitioning strategies. **Left:** Our main `DART-Flow` method. **Middle:** The grid-preserving `DART-Grid` method. **Right:** A visualization simulating the model's input for the `DART-Grid` method, created by reassembling the non-uniform patches into a uniform grid. In this view, high-density regions (e.g., the animal's body) appear magnified while low-density background areas are compressed. It is important to note that this is purely a visualization aid; the model's understanding of the original geometry is preserved through transformed positional embeddings and is not distorted.

## E   METHOD DETAILS

### E.1   ARCHITECTURAL OVERVIEW

To provide a comprehensive understanding of the DART framework, we present a detailed architectural diagram in Figure 12. The system operates as a dual-stream pipeline, ensuring that the content-aware partitioning is driven by high-level semantics while the final representation preserves low-level texture details.

The data flow proceeds through three key stages:

1. **Scoring Module (Control Stream):** As shown in the left column, the input image is processed by a lightweight, frozen feature extractor followed by a trainable scoring head. This module generates a low-resolution Score Map representing the spatial information density.

2. **Differentiable Quantile Function (Algorithm):** In the center column, the score map is converted into a probability distribution. We sequentially apply marginalization, Cumulative Distribution Function (CDF) computation, and the inverse CDF operation. This process is fully differentiable and outputs the adaptive partition boundaries (Adaptive Boundaries).

3. **Differentiable Resampling (Execution):** As depicted in the bottom row, the computed boundaries define a non-uniform grid. The **Differentiable Resampling** module takes two inputs: the generated grid coordinates and the *raw pixel stream* (which bypasses the scoring network via the dashed line). It extracts content-aware patches via bilinear interpolation. Finally, these patches are linearly projected and added with positional embeddings to form the final sequence of tokens.

This design ensures that while the *decision* of where to look is based on coarse semantics, the *content* of the tokens retains full high-frequency detail from the original image.

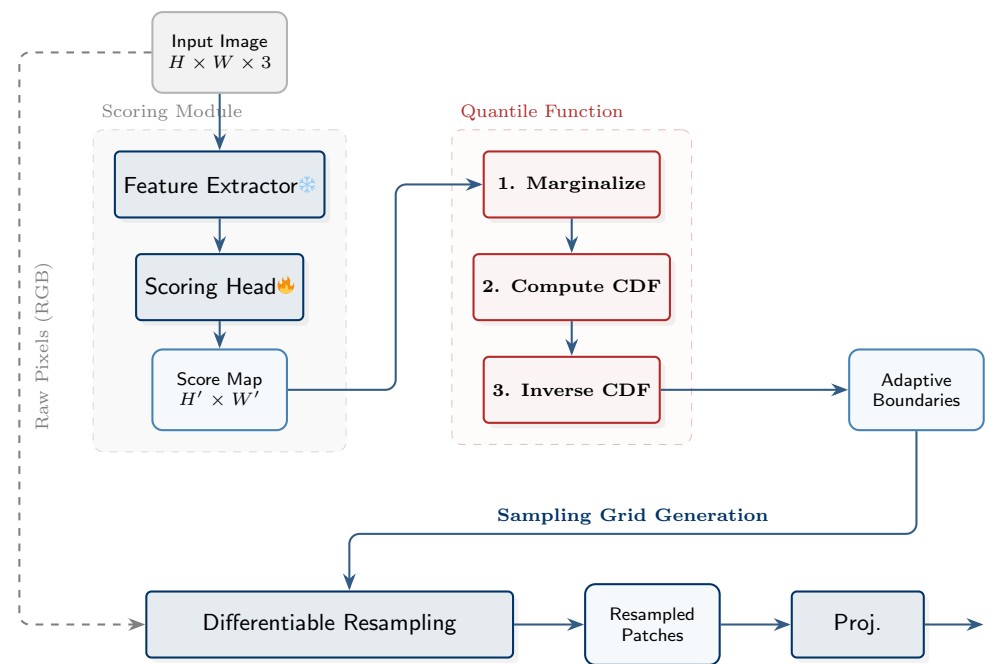

Figure 12: **Detailed Architectural Overview of DART.** The pipeline consists of a control stream (top) for boundary computation and a data stream (bottom/left bypass) for feature preservation.

### E.2 MATHEMATICAL FORMALISM OF THE DIFFERENTIABLE QUANTILE ALGORITHM

Our objective is to compute quantile boundaries from a discrete probability density function (PDF) within a fully differentiable framework. Given a 1D discrete probability distribution $S = \{s_0, s_1, \ldots, s_{L-1}\}$, where $\sum_{i=0}^{L-1} s_i = 1$ and $s_i \geq 0$, we aim to find a set of boundary points $\{x_1, x_2, \ldots, x_{K-1}\}$ corresponding to a set of target cumulative probabilities (quantiles) $Q = \{q_1, q_2, \ldots, q_{K-1}\}$.

**1. From Discrete PDF to Piecewise-Constant Function.** We model the discrete PDF $S$ as a piecewise-constant function $f(x)$ over the continuous interval $[0, L]$. For any integer index $i \in [0, L-1)$, the function's value is constant within the interval $[i, i+1)$:

$$f(x) = s_{\lfloor x \rfloor} \quad \text{for} \quad x \in [0, L) \tag{1}$$

**2. Constructing a Piecewise-Linear Cumulative Distribution Function (CDF).** The Cumulative Distribution Function (CDF), $C(x) = \int_0^x f(t)dt$, is derived from this PDF. This results in a continuous, piecewise-linear function. The value of the CDF at any integer point $j$ is the sum of all preceding probabilities:

$$C(j) = \sum_{i=0}^{j-1} s_i \quad (\text{with } C(0) = 0) \tag{2}$$

Within any interval $[j, j+1)$, the CDF increases linearly with a slope equal to $s_j$:

$$C(x) = C(j) + (x - j) \cdot s_j \quad \text{for} \quad x \in [j, j+1) \tag{3}$$

**3. Solving for Quantiles via CDF Inversion.** Finding the quantile boundary $x_k$ for a target probability $q_k$ is equivalent to solving the inverse problem $C(x_k) = q_k$. This is achieved in two steps:

1. **Locate the Interval:** For each target quantile $q_k$, we first identify the interval $[j, j+1)$ in which it falls. This is done by finding the first index $j$ such that the cumulative probability at

the end of the interval, $C(j + 1)$, is greater than or equal to $q_k$. That is, we find $j$ such that:

$$C(j) \leq q_k < C(j + 1) \tag{4}$$

In our implementation, this search is performed efficiently using a vectorized `argmax` operation over a boolean mask.

2. **Linear Interpolation:** Once the interval $[j, j + 1)$ is identified, we use the linear CDF equation for that segment to solve for the exact position of $x_k$:

$$q_k = C(j) + (x_k - j) \cdot s_j \tag{5}$$

By rearranging the terms, we obtain the analytical solution for $x_k$:

$$x_k = j + \frac{q_k - C(j)}{s_j} \tag{6}$$

Here, $C(j)$ is the cumulative probability mass before the start of the interval, and $s_j$ is the probability density within the interval.

**4. Differentiability.** The final expression for $x_k$ in Equation 6 is composed of differentiable operations (addition, subtraction, division, and summation). Although the interval selection step (`argmax`) is discrete and non-differentiable, it merely acts as a selector to determine which elements of $S$ participate in the final continuous calculation. As long as an infinitesimal change in the input distribution $S$ does not cause the index $j$ to change, the gradient of $x_k$ with respect to the elements of $S$ is well-defined and can be computed smoothly. This "differentiable almost everywhere" property is sufficient for end-to-end optimization using modern automatic differentiation frameworks like PyTorch.

### E.3 Algorithm Pseudocode

To enhance clarity and facilitate reproducibility, we present detailed pseudocode for our main topology-breaking **DART** (**DART-Flow**) algorithm in Figure 13. This pseudocode distills the core logic of our method, outlining the key sequential stages: generating an importance score map, performing adaptive row partitioning, reallocating the token budget globally across virtually flattened rows, and finally, executing the differentiable sampling of image content and positional information. It is important to note that while the description illustrates the conceptual flow, our actual implementation is highly parallelized and vectorized in PyTorch to fully leverage the computational power of modern GPUs.

To complement the mathematical formalism presented in Section E.2, we also provide detailed pseudocode for our differentiable quantile algorithm. Figure 14 describes the practical, highly-optimized implementation used in our work. Rather than a naive, iterative approach, it showcases a fully parallelized and vectorized solution that processes an entire batch of distributions simultaneously. The algorithm leverages efficient tensor operations like broadcasting to find interval indices and 'gather' to retrieve the necessary values for interpolation, completely avoiding slow, sequential loops. This design is critical for performance, as it fully exploits the parallel architecture of modern GPUs, ensuring the DART tokenizer remains a computationally lightweight component.

## F Extended Related Work: The Broader Efficiency Landscape

In the main text, we focused on methods directly related to dynamic token reduction and adaptive tokenization. Here, we broaden our scope to discuss the wider landscape of efficient Vision Transformer designs, categorizing them into static efficient architectures, dynamic depth mechanisms, and abstract token summarization. We highlight how DART occupies a unique and complementary niche within this ecosystem.

**Static Efficient Architectures.** A significant body of work focuses on designing lightweight, static backbones, often utilizing Neural Architecture Search (NAS) to optimize for specific hardware constraints. Hybrid Architectures like MobileViT Mehta & Rastegari (2022) and EdgeViT Pan et al. (2022b) combine CNNs with Transformers, while Pure Efficient ViTs like LeViT Graham et al.

```python
# Inputs:
#  x:            Input image tensor, shape (B, C, H, W)
#  scoring_net: A lightweight network to generate importance scores
#  pos_embed:   The original positional embedding defined on a uniform grid
#  N_h, N_w:    Target number of patch rows/columns (e.g., 14)
#
# Outputs:
#  final_tokens: The final sequence of tokens after dynamic sampling

def DART_tokenizer(x, scoring_net, pos_embed, N_h=14, N_w=14):
    N_total = N_h * N_w

    # 1. Generate 2D Probability Distribution from scores
    score_map = scoring_net(x)
    # ... (Normalization steps: sigmoid, per-sample sum to 1)
    prob_map_2d = normalize_scores(score_map)
    initial_pdf = prob_map_2d.flatten(1)

    # 2. Stage 1: Compute y-axis marginal and solve for horizontal boundaries
    pdf_2d_view = initial_pdf.view(-1, H_score, W_score) # Reshape for marginal
    marginal_prob_y = pdf_2d_view.sum(dim=2)
    cdf_y = torch.cumsum(marginal_prob_y, dim=1)
    y_boundaries = inverse_cdf(cdf_y, num_quantiles=N_h)
    row_heights = y_boundaries.diff(prepend=0)

    # 3. Stage 2: Virtually flatten rows and solve for all token boundaries
    resampled_pdf = resample_1d_by_grid(initial_pdf, grid=row_heights)
    resampled_pdf /= resampled_pdf.sum(dim=-1, keepdim=True)

    cdf_resampled = torch.cumsum(resampled_pdf, dim=1)
    final_edges = inverse_cdf(cdf_resampled, num_quantiles=N_total)

    # 4. Perform dynamic patch sampling from the original image
    patches = dynamic_image_patch_sample(x, row_heights, final_edges, (16, 16))

    # 5. Project patches and add transformed positional embeddings
    tokens = proj(patches)
    # ... (Positional embeddings are also resampled based on the new grid)
    final_pos_embed = transform_pos_embed(pos_embed, row_heights, final_edges)
    final_tokens = tokens + final_pos_embed

    return final_tokens
```

Figure 13: Pseudocode for our main **DART** (topology-breaking) tokenizer.

```
1242   # Function: inverse_cdf
1243   # Computes quantile boundaries from a batch of PDFs in a parallelized manner.
1244   #
1245   # Inputs:
1246   #  pdf:      A batch of probability density functions, shape (N, L)
1247   #  p_values: A 1D tensor of K target quantile probabilities, e.g., [0.25, 0.5, 0.75]
1248   #  eps:      A small constant to prevent division by zero (e.g., 1e-8)
1249   #
1250   # Outputs:
1251   #  quantiles: The computed quantile boundaries, shape (N, K)

1252   def inverse_cdf(pdf, p_values, eps):
1253       N, L = pdf.shape
1254       K = p_values.shape
1255
1256       # 1. Compute Cumulative Distribution Function (CDF) in parallel for the batch.
1257       cumsums = torch.cumsum(pdf, dim=1) # Shape: (N, L)
1258
1259       # 2. Find the interval index for each quantile in parallel.
1260       #    This is done by comparing each CDF against all p_values using broadcasting.
1261       mask = (cumsums.unsqueeze(-1) >= p_values.view(1, 1, -1)) # Shape: (N, L, K)
1262       j_indices = torch.argmax(mask.int(), dim=1)               # Shape: (N, K)
1263
1264       # 3. Gather all values required for interpolation in parallel using the indices.
1265       #    `gather` selects elements from the source tensor based on the index tensor.
1266       prev_j = torch.clamp(j_indices - 1, min=0)
1267       prev_area = torch.gather(cumsums, dim=1, index=prev_j)
1268       #    Handle the edge case where the index is 0.
1269       prev_area = torch.where(j_indices == 0, 0.0, prev_area)
1270
1271       pdf_val = torch.gather(pdf, dim=1, index=j_indices)
1272
1273       #    The start of the interval is simply the index j.
1274       edge_val = j_indices.float()
1275
1276       # 4. Perform parallel linear interpolation using the gathered tensors.
1277       #    The formula is applied element-wise across the batch and quantiles.
1278       quantiles = edge_val + (p_values.unsqueeze(0) - prev_area) / (pdf_val + eps)
1279
1280       return quantiles
```

Figure 14: Pseudocode for the parallelized, differentiable quantile computation algorithm ('inverse_cdf').

(2021) and EfficientFormer Li et al. (2022) optimize macro-architectures for latency. However, these specialized designs diverge from the standard uniform ViT architectures that currently dominate the landscape of large-scale foundation models (e.g., CLIP, SigLIP). Adopting a specialized efficient backbone often limits scalability and precludes the use of the rich, large-scale pre-training ecosystem available to standard ViTs. In contrast, DART is designed as a modular enhancement for the dominant uniform paradigm. It allows standard, scalable backbones (like DeiT and Vim) to become efficient while retaining their architectural compatibility with massive pre-trained foundation models.

**Dynamic Depth and Early Exiting.** Another prominent direction for efficiency is reducing the computational depth rather than the spatial width. Early-Exit methods (e.g., DynamicViT's early exit variants) allow "easy" samples to bypass deeper layers of the network once a confidence threshold is met. Layer Skipping approaches adaptively determine which layers to execute for a given input. These methods reduce compute by skipping *layers*, whereas DART reduces compute by processing fewer *tokens* per layer. Methodologically, DART is orthogonal to these approaches. The spatial

redundancy reduction of DART could theoretically be combined with the depth redundancy reduction of early-exit mechanisms to achieve compounded efficiency gains.

**Global Token Summarization.** Methods like TokenLearner Ryoo et al. (2021) and Perceiver Jaegle et al. (2021) adopt an attention-based approach to learn a small set of abstract latent tokens (e.g., 8-16 tokens) that summarize the global context of an image or video. While highly efficient for classification tasks, these methods typically perform weighted spatial pooling, aggregating information from disjoint regions into single abstract vectors. This abstraction process inherently sacrifices the grid-like spatial topology required for location-sensitive tasks. DART, conversely, preserves the spatial structure through region-based partitioning and coordinate-transformed positional embeddings.

## G USE OF LLM

In the preparation of this manuscript, we utilized Large Language Models (LLMs) as a writing assistant. The role of the LLM was strictly limited to improving the clarity, flow, and grammatical correctness of the text. Specific tasks included rephrasing sentences for better readability, correcting spelling and grammatical errors, and ensuring a consistent and professional tone. The authors have carefully reviewed and edited all text and take full responsibility for the final content of the manuscript.

