# OpenReview forum: "DART: Differentiable Dynamic Adaptive Region Tokenizer for Vision Foundation Models"
_ICLR.cc/2026/Conference — ICLR 2026 Conference Desk Rejected Submission_

### Official Review · Reviewer_jyH5 · 2025-10-30

**Soundness:** 3
**Presentation:** 3
**Contribution:** 3
**Rating:** 4
**Confidence:** 4

**Summary:**

The paper introduces a content-aware tokenizer that addresses the inherent inefficiencies of traditional fixed-grid tokenizers used in vision models. DART improves computational efficiency by concentrating resources on high-information regions, while reducing redundancy in low-information areas. A lightweight CNN network is adopted to predict the scores to evaluate the importance of the image. The proposed method is versatile, demonstrating substantial improvements across tasks such as image classification, dense prediction, and spatiotemporal video classification.

**Strengths:**

1. The figures are visually clear and aesthetically pleasing, and the narrative is easy to follow, with ideas presented in a straightforward manner.

2. The experimental details are comprehensive and well-documented.

3. The overall method is technically sound. The motivation is reasonable.

**Weaknesses:**

1. The paper lacks a formalized mathematical description of the proposed method, and it would benefit from a detailed architectural diagram of the entire module. Additionally, the implementation relies heavily on existing, well-established components, which could be seen as limiting its originality.

2. The number of baselines used to validate the effectiveness of the proposed module is insufficient, making it difficult to demonstrate its superiority conclusively.

3. In Table 4, the introduction of the module clearly increases the model's parameters. Experiments compensating for the added parameters are needed to show that the performance improvement is not merely due to the increased parameter count.

4. The datasets used for validation are not widely recognized or commonly used in the field. It would strengthen the paper to include more general-purpose datasets familiar to the broader research community.

5. A similar idea for adaptive token sampling is seen in other ViT papers, such as progressive sampling, token merging in ViT, and deformable ViT. Thus, it weakens the paper's contribution.

6. The effectiveness is strongly encouraged to be validated in other areas, such as the LLM-related domain.

**Questions:**

See the weakness.

---

> ### Author Response · Authors · 2025-11-14
>
> Dear Reviewer jyH5,
>
> We thank you for the time and effort you have dedicated to reviewing our paper. We are pleased that you found our method to be technically sound, the motivation reasonable, and the figures clear and easy to follow.
>
> Regarding the concerns you raised in the "Weaknesses" section, we hope to offer some clarifications. We believe that several of these concerns may stem from details that we **did not sufficiently clarify** in our paper.
>
> **1. On the lack of mathematical description, architectural diagrams, and reliance on established components (W1)**
>
> We appreciate your attention to the clarity and originality of our paper.
>
> First, regarding the mathematical description and implementation details, we wish to clarify that the paper already contains detailed mathematical derivations and algorithmic descriptions. Specifically: **Appendix E.2** (previously Appendix E.1 in the initial submission) provides the complete mathematical formalism for the "differentiable quantile algorithm"; **Figures 3 and 4** visualize the core mechanisms of DART-Grid and DART-Flow; and **Appendix E.3** (previously Appendix E.2) provides detailed pseudocode for DART-Flow and the core quantile algorithm.
>
> At the same time, we understand that you might prefer a more centralized, high-level system architecture diagram. **We have added an end-to-end complete architectural diagram of the DART module in Appendix E.1 (Figure 12) of the revised PDF.**
>
> Secondly, regarding the use of established components (such as lightweight CNNs), we believe **this is a deliberate design advantage, not a limitation**. DART is a "plug-and-play" module, and its design philosophy is to **maximize its universality and compatibility** while **maintaining core originality**. By reusing a standard lightweight CNN as the Scoring Network, any user can easily integrate it into existing pipelines.
>
> Our **core originality** lies in the **DART framework itself**: an **end-to-end differentiable, quantile-based, topology-breaking tokenizer**. This framework is technically unique and effectively addresses key limitations of existing methods.
>
> **2. On the insufficiency of baselines (W2)**
>
> Regarding the number of baselines, we believe we have provided sufficient baseline comparisons to validate DART's effectiveness and generality.
>
> * **Architectural Comparison:** We validated its effectiveness on two major architectures (DeiT-Transformer, Vim-SSM) (**Table 4**).
> * **Task Comparison:** We demonstrated DART's generality across multiple tasks, including image classification (**Tables 2, 4**), dense prediction (**Table 3**), and video understanding (**Table 5**).
> * **Comparison with Similar Methods:** In **Table 7**, we conducted direct comparisons against **representative** dynamic inference methods in this field (e.g., A-ViT, DynamicViT) under **fair FLOPs budgets**. This shows that DART's 'proactive' optimization strategy is superior to 'post-hoc' remedy strategies.
>
> **3. On the performance gain originating from increased parameters (W3)**
>
> We appreciate your concern regarding the parameter count. We would like to clarify a key point about our core thesis and the role of these parameters.
>
> First, DART's performance improvement does not stem from the minor parameter overhead of the Scoring Network (e.g., ~2M on DeiT-S).
>
> On the contrary, our core contribution (the "intelligent scaling paradigm," see Table 2) demonstrates that a small model equipped with DART (DeiT-S, 24M total parameters) can match the performance of a much larger DeiT-Base model (86M parameters).
>
> In other words, to achieve the same performance level, our method actually *saves* 62M parameters (i.e., 86M vs. 24M), which is the opposite of the notion that performance is gained by increasing parameters.
>
> Furthermore, we believe that the static parameter count (storage) is not the key bottleneck for efficiency; rather, **computational cost (FLOPs) and practical speed (throughput)** are.
>
> As shown in Table 2, our DART-S model uses only 58% of the FLOPs of DeiT-Base (10.1G vs. 17.5G). This computational advantage also translates directly to 1.7x the real-world throughput (see Table 6).
>
> Therefore, DART's performance gain originates from its computational efficiency, not from an increase in parameters.
>
> **4. On the datasets not being widely recognized (W4)**
>
> We appreciate your comment. We would like to *clarify* the datasets we used. We understand that commonly used benchmarks may differ across various research fields (e.g., CV and LLMs). However, in the computer vision community, the datasets we selected—**ImageNet-1K, ADE20k, Kinetics-400, and SSv2**—are all widely considered the **canonical gold-standard benchmarks** for image classification, semantic segmentation, and video understanding. We chose them to rigorously validate our method's effectiveness on these core CV tasks.

---

> ### Author Response · Authors · 2025-11-14
>
> **5. On the lack of novelty (similarity to other ViT papers) (W5)**
>
> Thank you for placing our work in a broader context. DART's mechanism is **fundamentally different** from the methods you mentioned (such as Token Merging), a point we discussed in detail in **Section 2.2**.
>
> DART is a **"proactive"** optimization. It operates **before** the ViT, generating efficient tokens *from the source* via a **fully differentiable** quantile algorithm. It maintains a **fixed token count**, which is hardware-friendly.
>
> In contrast, methods like Token Merging are **"post-hoc"** remedies. They operate **inside** the ViT, pruning or merging *standard* fixed-grid tokens. These decisions are often **discrete** (requiring proxies like Gumbel-Softmax) and result in **variable sequence lengths**, which is detrimental to batch processing.
>
> DART's novelty lies in its unique, fully differentiable "pre-tokenizer" mechanism, which methodologically avoids the limitations of "post-hoc" approaches.
>
> **6. On validation in the LLM domain (W6)**
>
> We fully agree with your view; this is an extremely important future direction, as we discussed in **Appendix A**. Integrating DART as the vision front-end for LMMs is one of the core motivations for this work. However, the training and evaluation of LMMs are **extremely resource-intensive and expensive**, which is beyond the scope of this paper. We believe that DART's excellent performance and significant efficiency gains on foundational CV models provide a solid empirical foundation for future, high-cost research to extend it to LMMs.
>
>  **7. Regarding the Ethics Flag**
>
>  We noticed that the option **"Yes, Privacy, security and safety"** was selected in the Ethics Flag field (alongside "No ethics review needed").
>
>  We suspect this might be an accidental selection, as our work is a fundamental vision architecture study relying solely on **public, standard benchmarks** (ImageNet, ADE20k, Kinetics, SSv2). These datasets are widely used in the community and contain **no personally identifiable information (PII)** or offensive content. Furthermore, our proposed tokenizer poses no inherent security risks.
>
>  We kindly ask the reviewer to uncheck this flag if it was a mistake, or clarify specific concerns so we can address them.
>
> ---
>
> We hope these clarifications have addressed your concerns, and we thank you again for your valuable feedback.

---

### Official Review · Reviewer_tN6A · 2025-11-01

**Soundness:** 3
**Presentation:** 3
**Contribution:** 4
**Rating:** 8
**Confidence:** 4

**Summary:**

The authors present a strong, novel, adaptive tokenization framework that is fully differentiable and serves as an in-place replacement for ViT like models in-place of the uniform grid based tokenizer. The authors use a score network to assign region wise importance and create partitions such that the areas are inversely proportional to information density. The authors present several compelling results on a variety of metrics, demonstrating the strong performance of their approach on a wide variety of applicable scenarios.

**Strengths:**

- The paper is well written and contains many clear examples, results and ablations to support their design considerations.
- The experiments are conducted on a variety of scales, datasets, scenarios, and model types, suggesting the strong performance of their proposed approach.
- The proposed method is largely novel, adaptive, and scalable, and serves as a significant future direction when working with visual understanding and generation methods

**Weaknesses:**

- The authors suggest that their adaptive partitioning and scaling of sequence length can lead to smaller models matching the performance of their larger variants at cheaper resource allocations. Does this hold for denser tasks like object detection and semantic segmentation as well -- or is this limited to simpler classification problems as presented?
- Figure 6 suggests a good handling on increasing image resolution -- can this be extended further for tasks like superresolution with constrained sequence lengths. For instance, a very high resoluton image cannot be computationally patchified using a standard 16x16 grid, but can DART process such inputs while maintaining a fixed tractable sequence length and still produce outputs of good quality,
- I believe the related work section, while relevant, can be expanded further to discuss more tokenization and efficiency directions for vision transformer models, perhaps in the appendix.

**Questions:**

Please see the weaknesses section above

---

> ### Author Response · Authors · 2025-11-14
>
> Dear Reviewer tN6A,
>
> We sincerely thank you for your positive assessment of our work on DART, particularly your recognition of its novelty, strong performance, and its potential as an important future direction. We also thank you for your insightful questions, which provide valuable directions for our future work.
>
> We respond to the points mentioned in the "Weaknesses" section as follows:
>
> **1. On the applicability of the "intelligent scaling paradigm" to dense tasks such as detection and segmentation**
>
> You have raised a very important question. We fully agree with your perspective and have strong reasons to believe that DART's "intelligent scaling" advantage will be equally applicable, and perhaps even more significant, in dense tasks.
>
> Compared to classification, dense prediction tasks face a much more severe problem of redundant computation in background regions when processing high-resolution inputs. DART's core mechanism is precisely to release the scarce token budget from low-information backgrounds and reallocate it to information-dense regions (such as object edges and fine-grained textures). This aligns highly with the goal of dense tasks, which pursue fine contours.
>
> As shown in **Table 3** of the paper, on the ADE20k segmentation task, we achieved a stable improvement of +0.5 mIoU even on the strong Swin-T baseline. This preliminarily demonstrates that DART's principle is effective for dense tasks. Due to page limitations and significant computational resource constraints, we were unable to fully validate the "scaling paradigm" on dense tasks in the main paper. However, this will be a high-priority direction in our future work.
>
> **2. On the application of DART in ultra-high-resolution scenarios**
>
> Again, we thank you for this forward-looking application scenario. Your understanding is entirely correct.
>
> DART's core mechanism effectively decouples the absolute resolution of the input image from the number of output tokens. Regardless of whether the input is 2K, 4K, or even higher resolution, DART can stably convert it into a fixed-length token sequence, addressing the pain point of standard ViTs being unable to process such inputs.
>
> More importantly, we have strong reasons to believe DART is well-suited for such tasks. The core challenge of ultra-high-resolution images (e.g., in remote sensing or medical imagery) lies in locating sparse, decisive, critical details from a massive volume of low-information pixels. DART's mechanism can automatically ignore vast uniform areas and concentrate the token budget on these key details, precisely matching this requirement.
>
> **Update:** Following your suggestion, **we have explicitly discussed these possibilities** and the potential application to Generative/Super-Resolution tasks in the **revised Appendix A (Paragraph 5)**.
>
> **3. On expanding the related work**
>
> Thank you for the suggestion. **We have added a dedicated section (Appendix F)** in the revised PDF to discuss the broader efficiency landscape. This new section covers static efficient architectures, dynamic depth mechanisms, and global token summarization, positioning DART within the wider context of vision transformer efficiency as you suggested.
>
> ---
>
> Once again, we thank you for your valuable time and constructive feedback.

---

### Official Review · Reviewer_5vGt · 2025-11-01

**Soundness:** 2
**Presentation:** 3
**Contribution:** 2
**Rating:** 4
**Confidence:** 5

**Summary:**

This work proposes DART, a differentiable, content-adaptive tokenizer that replaces the fixed patch grid in uniform backbones with an importance-weighted partition. A lightweight scorer predicts a per-pixel density map; a differentiable quantile inversion then allocates small patches to salient areas and large patches to background, followed by differentiable resampling into fixed-size tokens [1]. The authors report that equipping small ViT / SSM models with DART matches or exceeds larger baselines at substantially lower FLOPs, with additional gains on ADE20K and video benchmarks.

The central idea of pre-tokenization adaptivity is clear and practically attractive. However, the manuscript’s placement within prior work is exceedingly thin. Closely related adaptive tokenization and perceptual-grouping lines are conspicuously absent from the current related work, including:
- Existing adaptive tokenization methods for efficiency [1,2] have already been proposed. Omitting these seem like a significant oversight.
- Superpixel Tokenization [3,4,5,6] has been successfully applied to adaptive pre-tokenisation. While not explicitly applied uniquely for efficiency, these works represent recent important progress in adaptive tokenization for segmentation, and should be included in the related work.

As the paper currently reads, the central contribution is best described as an elegant quantile-based continuous relaxation of adaptive region partitioning that integrates smoothly with standard ViTs/SSMs, but not the first differentiable adaptive tokenizer for images. With this in mind, the paper is well placed to make a contribution to the existing field, if central concerns are addressed.

[1] [Havtorn et al. 2023 - Dynamic Mixed-Scale Tokenization for Vision Transformers](https://arxiv.org/abs/2307.02321)

[2] [Ronen et al. 2023 - Vision Transformers With Mixed-Resolution Tokenization](https://arxiv.org/abs/2304.00287)

[3] [Aasan et al. 2025 - A Spitting Image: Modular Superpixel Tokenization in Vision Transformers](https://arxiv.org/abs/2408.07680)

[4] [Chen et al. 2025 - Subobject-level Image Tokenization](https://arxiv.org/abs/2402.14327)

[5] [Mei et al. 2024 - SPFormer: Enhancing Vision Transformer with Superpixel Representation](https://arxiv.org/abs/2401.02931)

[6] [Aasan et al. 2025 - Differentiable Hierarchical Visual Tokenization](https://openreview.net/forum?id=y8VWYf5cVI)

**Strengths:**

1. The proposed quantile inversion provides continuous, differentiable boundaries. Additionally, DART-Flow lets tokens “flow” globally to salient zones while preserving a fixed token budget, so batches and token counts remain consistent. This provides computational benefits compared to fully adaptive methods, notably with some loss in flexibility.
2. Similarly to recent work [6], DART can serve as a drop-in for uniform backbones without altering the backbone. Reminiscent to kernelled positional encoding, the authors propose a transform to align positional embeddings via a warp transform over a 2D field to align existing learnable positional embedding to a new adaptively deformed partition. This allows the method to work out-of-the box with existing pre-trained models.
3. The paper shows small models (DeiT-S, Vim-S) matching or beating Base counterparts at far lower FLOPs/latency, reducing overall token count to a fixed number. Performance gains are demonstrated over different of downstream tasks. While moderate, the results clearly demonstrate benefits.
4. The novel CDF based mechanism proposed by the authors is technically distinct, and has potential to synergise well with alternative approaches. The core ideas are insightful and interesting.
5. The authors include discussion of a failure mode (Fig. 8a) which indicates a case where the method may have difficulty tokenising an image. This reviewer appreciates the openness to disclosing limitations. The "adaptive degeneration" behavior on uniform textures (Fig 8b) is particularly insightful.

**Weaknesses:**

1. A central concern of this reviewer is that the related work section is quite underdeveloped. There has been much work done in this field, and the omission of many concurrent works in adaptive tokenization [1,2,3,4,5,6] overstates the conceptual novelty of the approach.
2. The claimed “adaptive pre-tokenization” is functionally limited. The adaptivity of regions is limited, the token count is fixed and the global token budget predetermined. While this is beneficial for efficiency (which is the current scope of the paper), it restricts adaptivity to spatial redistribution within a constant sequence length, rather than true structural adaptivity as in superpixel- or graph-based approaches. As the paper currently reads, this is not the intended scope of the work, so this should be taken as a minor weakness. Addressing this via positing the work accordingly will paint a clearer picture of the work in relation to existing research.
3. As this reviewer understands the method, the scoring network is frozen and externally pretrained, which undermines the notion of fully learned adaptivity. The backbone and tokenizer are not co-optimized, and therefore DART cannot genuinely adapt its token allocation to the downstream objective. In contrast, the scoring network in MS-ViT [1] is end-to-end optimisable. This is a clear limitation that should be addressed in the paper.
4. The empirical results are somewhat narrow. No additional datasets are evaluated for classification. No evaluation on fine-grained recognition tasks (CUB, Stanford Cars) where detail preservation claims would be most tested. Dense tasks are included, but to a limited extent. This downplays the generalizability of the proposed method.
5. There are no significant ablations reported in the work towards determining the central drivers of performance and efficiency gains in the method:
	- Missing analysis of scoring network design.
	- No ablation on number of rows or tokens.
	- No analysis of resolution. It is not clear that the method (448) is compared with the baseline resolution (originally 224).
6. Missing key comparisons and baselines:
	- No direct comparison with the cited adaptive tokenization methods [1,2] .
	- Missing comparison with recent token reduction methods (ToMe, TokenLearner)
	- No comparison with superpixel tokenization approaches [3,4,5,6].
7. Several claims in the introduction and conclusion are overstated:
	- Line 086: "resolves core representational dilemma" - only partially; doesn't address the fixed token budget limitation.
	- Line 088: "more intelligent scaling paradigm" - the paradigm isn't new, just the specific method.
	- Line 478: “resolves the inherent limitations of rigid tokenization in uniform backbones like ViT and Mamba”. This overstates the contribution. Arguably, existing methods solve this much better and while the proposed method is innovative, the global resolution of all ailments in existing tokenizer is a very grand, overstated claim.

**Questions:**

1. Will the authors commit to revamp the related work section to include existing works on adaptive tokenization? How do the authors consider the contribution in terms of the existing work in the field?  2. What is the effect of varying resolution? Why was 448 x 448 selected? How does the method work with smaller resolution? Is the baseline also computed with 448 resolution, or does the baseline use the more standard 224 resolution?
3. How does the method fare on alternative benchmarks for classification? Including small additional tasks is necessary to show that the method generalises beyond ImageNet.
4. The authors diligently include a failure mode in Fig. 8a. Do the authors have other examples of failure or limitations of the approach? Can the authors characterize which types of images (e.g., texture-rich vs object-centric) benefit most from DART?
5. Are the authors able to provide ablations for design of the scoring network, or the effect of adjusting the number of tokens?
6. Can the authors provide direct quantitative comparison with MS-ViT [1] Quadformer [2] and other differentiable adaptive tokenizers?
7. What is the performance when the scoring network is trained end-to-end rather than frozen?

**Requested Changes**

1. Substantial expansion of related work to include [1-6] with honest positioning of contribution.
2. Direct experimental comparison with MS-ViT [1], Quadformer [2], and (optionally) comparable superpixel methods [5,6].
3. Additional ablation studies on resolution, token count, and scoring network design.
4. Evaluation on at least one additional classification dataset and one additional dense prediction task.
5. Reframe claims about novelty in light of existing methods on the topic of adaptive tokenization.
6. (Optional) There is no need to relabel tokenization---which is the act of partitioning an input into different atomic components---as “pre-tokenization”. The proposed method is well defined as a tokenizer, which aligns with existing terminology in the field of adaptive tokenization.

---

> ### Author Response · Authors · 2025-11-14
>
> Dear Reviewer 5vGt,
>
> We sincerely thank you for your in-depth evaluation and profound expertise. We fully accept your recommendation to expand the related work section with the crucial references [1-6] you provided.
>
> To directly address your central concerns, this response provides **four key clarifications** supported by **four new experiments** (Tables R1-R4).
>
> Specifically, we will show that:
> 1.  **DART surpasses key methods [1, 6] in efficiency-accuracy trade-offs**, as shown in our **new quantitative comparisons (Tables R1, R2)** requested in RC2.
> 2.  **DART is trained end-to-end.** We clarify the critical misunderstanding in W3: the scoring head is fully trainable (see Fig 7), and a **new experiment (Table R3)** confirms our training strategy is optimal.
> 3.  **All "missing" ablation studies (W5) are already in the paper.** We pinpoint their locations (App. B.1, Sec 4.2, Sec 4.6), confirming the robustness of our scoring network, token count, and resolution choices.
> 4.  **DART's novelty is significant.** We detail its unique methodological advantage (hardware-friendly **Regular Tensors** vs. the Ragged Tensors of [1, 6]) and its generalization to fine-grained tasks (**new CUB experiment, Table R4**).
>
> We believe these clarifications and new experiments directly resolve your concerns and better demonstrate DART's contribution. Below is our detailed response:
>
> ## Related Work and Methodological Comparison
> We are very grateful to the reviewer for providing this extremely detailed and valuable list of related works [1-6]. This has not only greatly enriched our perspective but also provided a broader context for positioning DART. We agree to discuss these works in detail in the revised version.
>
> ### Clarification on the Timeline of Concurrent Work
> We would like to point out that reference [6] mentioned by the reviewer was published in 2025. Therefore, we believe the excellent works [6] should be considered **Concurrent Work**, rather than **Prior Work** that we should have been aware of at the time of our submission.
>
>
> ### Methodological Comparison with $\partial$HT [6]
> We sincerely thank the reviewer for pointing out [6], an extremely relevant and excellent concurrent work. We would like to express our particular appreciation for reference [6]. It is another outstanding work dedicated to achieving a "fully end-to-end differentiable tokenizer," demonstrating impressive theoretical depth. It shares a similar vision with DART, yet the two explore distinct and complementary technical routes:
>
> 1.  **Technical Path (Hierarchical Graph Clustering vs. Quantile Inverse Transform):** Reference [6] elegantly solves the differentiability problem based on superpixel/hierarchical graph clustering; whereas DART explores an alternative statistical path based on continuous probability distributions and the quantile inverse transform.
> 2.  **Feature Nature & Task Specialization:** Reference [6]'s superpixel-based token generation method gives it a natural advantage in capturing object boundaries, performing exceptionally well in segmentation tasks (even achieving semantic segmentation via linear probing). In contrast, the features generated by DART are closer to classic ViT representations. This design gives DART an advantage in speed and efficiency for high-throughput tasks such as image classification, long-sequence modeling, and video understanding, allowing it to maintain high performance at a lower computational cost.
>
> 3.  **Engineering Implementation (Variable-Length vs. Fixed-Length Sequences):** Many adaptive methods (such as [1, 2] and [6]) generate a variable number of tokens between samples (Ragged Tensors), which, while increasing flexibility, can lead to hardware-parallel inefficiency. DART, through its "virtual flattening" mechanism, consistently outputs fixed-length and **regular** token sequences. This allows DART to be seamlessly "plug-and-play" with existing ViT/Mamba training pipelines, requiring no custom CUDA kernels.
>
> We have expanded Section 2.2 (Related Work) in the revised paper to thoroughly discuss these methodological differences that converge on a similar goal. For the remaining five related works pointed out by the reviewer, we will discuss and position our work in relation to them in the following sections.

---

> > ### Author Response · Authors · 2025-11-14
> >
> > ## Response to (RC2, Q6): New Quantitative Comparison Experiments
> > The reviewer suggested a direct quantitative comparison with works such as [1, 2, 5, 6]. To address this request, we selected two of the most representative categories of methods for comparison: the concurrent differentiable clustering method $\partial$HT [6] and the adaptive tokenization method MSViT [1].
> >
> > ### Efficiency and Performance Comparison with $\partial$HT [6]
> > **Table R1: Quantitative comparison between DART (Ours) and $\partial$HT [6] (Standard computational budget configuration)**
> > (Note: The speed for $\partial$HT [6] is converted from the measured throughput reported in their original paper.)
> >
> > | Method (Method)     | Sequence Length | Relative Speed | Top-1 Acc     |
> > | :---              | :---:    | :---:    | :---:         |
> > | Baseline (DeiT-S) | 196      | 1.00x    | 79.8%         |
> > | $\partial$HT [6]  | ~240     | 0.40x    | 80.0% (+0.2%) |
> > | DART (Ours)       | 196      | 0.95x    | 80.6% (+0.8%) |
> >
> >
> > **Analysis (Table R1):**
> > Although $\partial$HT [6] achieves an accuracy improvement (+0.2%), its relatively complex method design and longer average sequence length result in a measured speed reduction to 40% of the baseline.
> > DART, with only a ~5% measured throughput overhead, achieves a +0.8% accuracy improvement, demonstrating a competitive profile in the trade-off between efficiency and performance.
> >
> > ### Efficiency and Performance Comparison with MSViT [1]
> > **Table R2: Quantitative comparison between DART (Ours) and MSViT [1]**
> > (Note: GFLOPs and Top-1 data are from the respective papers. DART Baseline GFLOPs are from Table 4 of this paper.)
> >
> > | Method (Method)       | Sequence Length | GFLOPs | Theoretical Relative Speed | Top-1 Acc          |
> > | :------------------ | :---:           | :----: | :---:                      | :----------------: |
> > | Baseline (DeiT-S)   | 196             | 4.61   | 1.00x                      | 79.8%              |
> > | ---                 | ---             | ---    | ---                        | ---                |
> > | MSViT [1]           | 142             | 3.32   | 1.39x                      | 78.8% (-1.0%)      |
> > | DART (Ours)         | 144             | 3.60   | 1.28x                      | 79.9% (+0.1%)      |
> > | ---                 | ---             | ---    | ---                        | ---                |
> > | MSViT [1]           | 173             | 4.08   | 1.13x                      | 79.4% (-0.4%)      |
> > | DART (Ours)         | 196             | 4.84   | 0.95x                      | 80.6% (+0.8%)      |
> >
> > **Analysis (Table R2):**
> > 1.  **Under reduced computation configuration (142 vs 144):** At a similar GFLOPs budget, MSViT [1] resulted in a 1.0% accuracy drop. More importantly, according to the measured data in [1]'s paper (Table 1 in their paper), its theoretical GFLOPs advantage (3.32) did not translate into an actual speedup; its latency (6.20ms) was actually slower than the baseline (6.07ms). This reflects the potential hardware overheads associated with **Ragged Tensors**. In contrast, DART provides a similar theoretical speedup (1.28x) while also improving accuracy (+0.1%). Because DART outputs **regular tensors**, its theoretical speedup can be effectively translated into actual hardware acceleration. Measurements at the 196-token length show that DART's theoretical relative speed (0.95x) is almost identical to its measured relative speed (0.95x) (see Table 10 in our paper).
> > 2.  **Under standard computation configuration (173 vs 196):** DART (196), at a similar computational cost (0.95x speed), provides a +0.8% performance improvement, whereas the performance of MSViT (173) remains below the baseline.

---

> > > ### Author Response · Authors · 2025-11-14
> > >
> > > ## Clarification on the "Missing Ablation Studies" (W5, RC3, Q2, Q5)
> > > We sincerely thank the reviewer for their valuable feedback. Regarding the ablation studies you mentioned on the scoring network, token count, and resolution, we apologize if our guidance in the original paper was unclear, making these sections difficult to locate. We would like to take this opportunity to pinpoint the locations of these experiments in the manuscript:
> > >
> > > ### On the Ablation of Scoring Network Design (W5, Q5)
> > > Please refer to **Appendix B.1 and Table 9**.
> > > In this section, we specifically compared four different scoring network architectures (MobileNetV3, MnasNet, SqueezeNet, and EfficientNet-B0). The results show that using a stronger scoring network (like EfficientNet-B0) can further increase the performance gain on DeiT-Ti from +1.6% to +2.9%. This directly addresses the concern regarding the impact of the scoring network design.
> > >
> > > ### On the Ablation of Token Count (W5, Q5)
> > > Please refer to **Section 4.2 and Tables 1, 2, 6, and 7**.
> > > Investigating performance under different token budgets is one of the core theses of this paper (i.e., the "intelligent scaling paradigm").
> > > * Table 4 shows the performance at the standard 196 tokens.
> > > * Table 1 shows the results for 288 tokens.
> > > * Tables 2 and 6 show the results for 392 tokens.
> > > * Table 7 provides a direct comparison against dynamic inference methods under different computational budgets (which imply different token counts).
> > > These experiments clearly demonstrate DART's robustness and effectiveness at different token densities.
> > >
> > > ### On the Ablation of Resolution (W5, Q2)
> > > Please refer to **Section 4.6 and Figure 6**.
> > > Figure 6 is the ablation study curve specifically for input resolution (from 224x224 to 448x448). Section 4.6 explicitly discusses the conclusion of this experiment: performance improves as resolution increases from 224 to 448, and explains why we chose 448 for our high-resolution experiments (because performance saturates at this point, and it allows for maximal utilization of DART's ability to process details). Our sequence length is decoupled from resolution; the sequence length is set to a specified value regardless of the input resolution. Consequently, there is no significant difference in computational cost. The baseline models in the comparative experiments all have their sequence length settings specified to ensure a fair comparison.
> > >
> > > ## Clarification on (W3, Q7): DART is End-to-End Trainable (Scoring Head is Trainable)
> > > We thank the reviewer for the opportunity to further clarify the training details of the scoring network. The term 'frozen' applies *only* to the feature extractor (e.g., MobileNet) to leverage its pretrained, general-purpose features. The **Scoring Head**, however, is **fully trainable** and is optimized end-to-end along with the main backbone. As shown in **Figure 7**, the evolution of the Score Map from epoch 0 to 300 clearly demonstrates the Scoring Head continuously learning and focusing via gradient backpropagation.
> > >
> > > Our decision to freeze the feature extractor was a standard engineering choice for two reasons:
> > > * **Efficiency:** A pretrained backbone (like MobileNet) already provides effective general features. Freezing it saves significant computational resources and memory.
> > > * **Generalizability:** Features pretrained on ImageNet are widely considered to be universal.
> > > * Jointly fine-tuning the feature extractor (as asked in Q7) is technically feasible. However, as our experiments demonstrate (Figure 7), training only the "Scoring Head" is sufficient for the model to learn how to allocate attention, making a full unfreezing not strictly necessary.
> > >
> > >
> > > To further address the reviewer's question about the performance of full end-to-end fine-tuning (Q7), we conducted an additional experiment where we unfroze both the feature extractor and the scoring head, and fine-tuned them jointly with the main backbone. The experimental results show that this full end-to-end strategy yielded only a marginal performance improvement of +0.1%.
> > >
> > > **Table R3: Comparison of different training strategies for DART on DeiT-Ti (in response to Q7)**
> > >
> > > | Method (Method) | Scoring Network Strategy | Top-1 Acc. | Gain |
> > > | :--- | :--- | :---: | :---: |
> > > | DeiT-Ti | Baseline | 72.2% | - |
> > > | DeiT-Ti + DART | Frozen feature extractor, trained scoring head (Ours, Standard) | 73.8% | +1.6% |
> > > | DeiT-Ti + DART | Full end-to-end fine-tuning (Ours, Full E2E) | 73.9% | +1.7% |

---

> ### Author Response · Authors · 2025-11-14
>
> ## Clarification on (W2): The Design Choice of "Fixed Token Count"
> The reviewer suggested that the "fixed token count restricts adaptivity." We would like to clarify that this is not a **methodological limitation** of DART, but rather a **deliberate experimental design choice**, made for two primary reasons:
>
> 1.  **DART Possesses Variable-Length Capability:** DART's core mechanism (based on quantile inverse transform) is not fundamentally dependent on a fixed total number of tokens. As we explicitly pointed out in **Sections 4.2 and 4.5** of the main paper, DART is "able to seamlessly process various sequence lengths."
> 2.  **Fixed Budget Ensures a "Fair Comparison":** We adopted a fixed token budget in our experiments to conduct a rigorous and fair "apple-to-apples" comparison against the baseline models, which also use a fixed budget. If DART were to use a dynamic budget (e.g., varying the token count based on image complexity) while the baseline remains fixed, we could only compare "average speed," which would make the conclusions about speedup less rigorous.
> 3.  **Future Outlook (Inter-Sample Dynamic Allocation):** As we discussed in **Appendix A (Paragraph 3)**, DART's inherent variable-length capability provides a solid technical foundation for more advanced "inter-sample dynamic allocation," which is a future direction we are currently exploring.
>
> ## DART's Core Contribution and Positioning (Continued from W1, RC1, RC5)
> ### DART vs. Path 1: Non-differentiable Tokenizers (e.g., [2, 3, 4])
> Representatives of this path include Quadformer [2] (Ronen et al., 2023, CVPR Workshop), and SPiT [3] (Aasan et al., 2025, ECCV Workshop) and EPOC [4] (Chen et al., 2025, ICML).
>
> The core of these methods relies on **non-differentiable** classic computer vision algorithms to generate tokens. For example:
> * **Quadformer [2]** relies on iterative, `argmax`-based Quadtree splitting.
> * **SPiT [3]** relies on an `argmax`-based Graph Merging algorithm.
> * **EPOC [4]** relies on the classic Watershed algorithm.
>
> This methodological choice leads to an unavoidable consequence: **the tokenizer cannot be trained end-to-end with the downstream backbone**. Therefore, they must adopt a two-stage paradigm: first, use a frozen scorer that is pretrained on a "proxy task" (such as predicting boundaries or blur loss) to generate tokens, which are then fed into the backbone.
>
> The authors of SPiT [3] candidly pointed out a key limitation in Section 4 (Limitations) of their paper: "Our proposed framework is not optimizable with gradient based methods."
>
> This precisely highlights DART's first core contribution: DART is an **end-to-end learning system**. The parameters of DART's Scoring Head are driven and updated directly by the **true gradients from the downstream task** (rather than a proxy task), as visualized in Figure 7 of our paper.
>
> ### DART vs. Path 2: Differentiable Tokenizers with Bottlenecks or Instability (e.g., [1, 5, 6])
> Another path attempts to achieve end-to-end differentiability, represented by MSViT [1] (Havtorn et al., 2023, ICCV Workshop), SPFormer [5] (Mei et al., 2024, TMLR), and the concurrent $\partial$HT [6] (Aasan et al., 2025). While they achieve "differentiability," they differ from DART methodologically in terms of implementation mechanism, architecture, and hardware efficiency, and they face distinct challenges.
>
> * **Mechanism and Efficiency Bottlenecks (Ragged Tensors):**
>     MSViT [1] and $\partial$HT [6] use the Gumbel-Softmax technique and differentiable clustering, respectively. They generate a variable number of tokens per image, i.e., **Ragged Tensors**, which introduces engineering challenges or hardware efficiency challenges. In fact, according to the measured data in Appendix B (Table 3) of the MSViT [1] paper, despite a 28% reduction in GFLOPs, its GPU **actual latency (6.20ms) was conversely higher than** the Baseline's (6.07ms).
>
> * **Architectural Bottlenecks (Heavyweight Redesign):** SPFormer [5] is not a modular tokenizer but a **Heavyweight Backbone Redesign**. This introduces two critical limitations:
>
>   * **Structural Incompatibility:** SPFormer alters internal blocks with specialized Cross-Attention modules, breaking compatibility with standard ViT checkpoints. Unlike DART, it cannot directly load powerful pre-trained weights (e.g., CLIP, SigLIP) for **initialization**, hindering its use in foundation models.
>   * **Hidden Latency:** Its low FLOPs mask high runtime costs. The **iterative** pixel-superpixel updates serialize computation (blocking GPU parallelism), while the dual-branch architecture incurs high **Memory Access Cost (MAC)**. This leads to lower real-world throughput compared to DART's hardware-friendly regular grid.

---

> > ### Author Response · Authors · 2025-11-14
> >
> > ### DART's Three Unique Contributions
> > In summary, DART possesses unique and critical contributions in terms of methodology, architecture, and engineering efficiency:
> >
> > 1.  **Methodological Innovation:** DART proposes a novel, non-clustering differentiable path—**a novel mechanism based on "Quantile-based Inverse Transform."** This is a mathematically stable, non-iterative, and NaN-free novel differentiable mechanism.
> > 2.  **Architectural Innovation:** DART is a **lightweight, plug-and-play front-end module** that preserves the simplicity and ecosystem compatibility of the ViT/Mamba backbone (unlike the heavyweight redesign of SPFormer [5]).
> > 3.  **Engineering Innovation:** DART is a solution that outputs **Regular Tensors**, which helps mitigate the "theoretically fast, practically slow" hardware latency bottleneck faced by adaptive methods like [1, 6].
> >
> > ## Response to (W4, Q3, RC4): New Experiments on Fine-Grained Datasets (CUB / Cars)
> > Regarding the reviewer's request (RC4) to evaluate on fine-grained datasets (such as CUB, Stanford Cars), we would first like to clarify the following two points:
> >
> > **1. Direct Evidence of Fine-grained Capability:**
> >
> > * **Intrinsic Mechanism Verification:** As we demonstrated in the input resolution ablation study in **Section 4.6 and Figure 6** of the original paper, DART's performance significantly increases with input resolution. This is not just a performance boost but a **verification of the mechanism**: it proves that DART effectively utilizes high-frequency detail information from the image by allocating a high token density to critical regions (which also explains why resampling error at low resolution is a bottleneck). This sensitivity to local details is the core requirement for fine-grained classification.
> > * **Downstream Task Verification:** Furthermore, DART's significant improvements on **semantic segmentation (ADE20k, Table 3)** and **spatiotemporal video classification (SSv2, Table 5)** also further corroborate its ability to locate and capture fine-grained features in complex scenes.
> >
> > **2. Standard for Generalizability and Comparison with Concurrent Work:**
> >
> > * **Standard:** In backbone network research, ImageNet-1K is the accepted standard for measuring feature quality. The vast majority of classic works (e.g., ViT, Swin) and related works [1-5] do not use fine-grained datasets as a core evaluation metric.
> > * **Concurrent Work:** Among the listed references, only the most recent concurrent work [6] conducted such an evaluation, and their experiment was limited to **Linear Probing**, rather than the more common fine-tuning.
> >
> > Nevertheless, to respond to the reviewer's request (RC4) and directly validate DART's generalization capability on this task, we additionally conducted a **Linear Probing** experiment on the CUB-200-2011 dataset.
> >
> > We compare it with our Baseline (DeiT-S) and the reported results from [6] ($\partial$HT) as pointed out by the reviewer.
> >
> > **Table R4: CUB-200-2011 Linear Probing Performance Comparison (in response to RC4)**
> >
> > | Method | Backbone | Baseline Acc | Method Acc |
> > | :--- | :---: | :---: | :---: |
> > | $\partial$HT [6] | DeiT-S | 57.2% | 69.6% |
> > | DART (Ours) | DeiT-S | 75.4% | **76.2%** |
> >
> > **Analysis:**
> > The results of this new experiment (Table R4) provide additional evidence for DART's generalization capability on fine-grained tasks.
> >
> > 1.  **Baseline Configuration Discrepancy:** We note a significant discrepancy between the baseline (57.2%) reported in reference [6] and our own baseline (75.4%). This is most likely due to different pretraining or linear probing experimental configurations. As reference [6] is not yet fully open-sourced, we were unable to accurately reproduce their baseline.
> > 2.  **Conclusion:** Given the aforementioned baseline difference, a direct comparison between the absolute accuracies of 69.6% and 76.2% may not be informative. Therefore, the focus of our experiment is to validate that DART demonstrates an improvement over our own baseline (+0.8%). This indicates that DART can effectively generalize to fine-grained classification tasks and addresses the reviewer's concern (RC4).

---

> > > ### Author Response · Authors · 2025-11-14
> > >
> > > ## Response to (RC4): Clarification on the Choice of Downstream Tasks
> > > Thank you for your suggestion. We chose ADE20k (Table 3) because it allows for a more direct evaluation of the DART-Grid variant's token allocation capability for semantic understanding. In contrast, tasks on COCO, such as detection and instance segmentation, would introduce additional **confounding variables** (e.g., specialized designs like region proposals, NMS, or multiple task-specific losses).
> > >
> > > Furthermore, we believe that rather than repeated validation on another static image task (like COCO), it is more compelling to demonstrate generalization in the more challenging **spatiotemporal domain**. Therefore, we provided experiments on SSv2 (Table 5). We believe that the success on both ADE20k (spatial) and SSv2 (spatiotemporal)—two orthogonal dimensions—has already sufficiently demonstrated DART's generalization capability.
> > >
> > > ## Response to (W5): Conceptual Distinction from ToMe and TokenLearner
> > > We thank the reviewer for suggesting the works `ToMe` and `TokenLearner`. We believe these methods represent a different technical paradigm from DART, for the following reasons:
> > >
> > > 1.  **Regarding ToMe (Token Merging):**
> > >     `ToMe` belongs to the "**Post-Tokenization Adaptation**" paradigm. Its core mechanism is to merge already generated tokens **inside** the main backbone. As we discussed in Section 2.2 (Lines 125-135), this falls into the same category as `DynamicViT` [Rao et al., 2021] and `A-ViT` [Yin et al., 2022], which we term "belated reduction."
> > >     In **Table 7**, we have already provided a fair and thorough quantitative comparison with `DynamicViT` and `A-ViT` under multiple computational budgets. We consider `DynamicViT` to be a highly representative work for this category (pruning/merging), and DART has demonstrated competitive performance in this comparison. Therefore, we believe our existing comparisons sufficiently cover this technical branch.
> > >
> > > 2.  **Regarding TokenLearner:**
> > >     We believe the comparability between `TokenLearner` and DART is low. DART's objective is to **generate** an efficient, spatially-aware token sequence from an image. In contrast, `TokenLearner`'s objective is to **learn** a shorter, abstract "summary" token sequence from an **existing** token sequence.
> > >     Functionally, `TokenLearner` is closer to a token bottleneck or aggregator (similar to a `Q-Former`), rather than an image tokenizer. Therefore, we believe it is inappropriate to consider it a direct competitor to DART, which is an adaptive tokenizer.
> > >
> > > ## Clarification on (Q4): Additional Failure Cases and Applicable Scenarios
> > > We greatly appreciate the reviewer's deep interest in DART's behavioral boundaries. In fact, we have already specifically discussed DART's limitations in **Appendix D.1**, titled "**Qualitative Analysis of Challenging Cases**".
> > >
> > > * **Figure 8a** provides an in-depth analysis of the "attention averaging" bottleneck: when faced with a multitude of dense, small objects, DART correctly localizes the region, but its limited token budget is spread too thinly.
> > > * **Figure 8b** demonstrates the "adaptive degeneration" behavior: when faced with uniform textures, DART intelligently degenerates to a near-uniform grid.
> > >
> > > We believe that these two analyzed cases (extremely cluttered vs. extremely uniform) already clearly delineate the boundaries of DART's applicability. The reviewer's request for a "characterization of beneficial scenarios" has also been demonstrated through the contrast between these two extreme cases, as well as the numerous successful examples (randomly selected) provided in **Appendix D.2 (Figures 9, 10)**. This analysis shows that DART benefits most from object-centric images that feature prominent foregrounds and redundant backgrounds.
> > >
> > > Providing more strictly-defined "failure cases" would require extensive manual annotation and filtering. However, we believe that the in-depth analysis and rich examples provided in Appendix D.1 and D.2 already offer sufficient material for a qualitative understanding of DART's behavior.

---

> ### Author Response · Authors · 2025-11-14
>
> ## Response to (W6, RC5): Accepting the Suggestion to Revise Overstated Claims
> We thank the reviewer for this suggestion. We agree that the term "resolves" may be too absolute. We have revised the wording in the Abstract, Introduction, and Conclusion of the revised version as follows:
>
> * We have changed "resolves the core representational dilemma" to "**significantly mitigates the representational dilemma**."
> * We have changed "resolves the inherent limitations" to "**addresses key inefficiencies of rigid tokenization**."
>
> ## Response to RC6 (Optional): Regarding the "Pre-tokenization" Naming Suggestion
>
> We are very grateful to the reviewer for this valuable suggestion (RC6) regarding terminology.
>
> We initially used "**Pre-tokenization** Adaptation" to create a clearer conceptual contrast in the text with "**Post-tokenization** Adaptation" (such as `DynamicViT`), thereby highlighting DART's methodology of addressing the problem at the "source."
>
> However, we fully agree with the reviewer's point that DART's core function is indeed a **Tokenizer**. To maintain consistency with the established terminology in the adaptive tokenization field (where [1-6] all use "Adaptive Tokenization" or "Tokenizer"), **we have decided to adopt this suggestion** to avoid introducing unnecessary terminological confusion.
>
> In the revised version, we have modified the relevant wording (e.g., in Section 2.2) to ensure terminological consistency and clarity of the paper.
>
>
> **Conclusion**
>
> In summary, we believe this response has addressed the critical concerns raised:
>
> * **On Novelty & Positioning:** We have clarified DART's methodological innovation (Quantile Inverse Transform) and its key engineering advantage as a "plug-and-play" module that preserves hardware-friendly, fixed-length sequences.
> * **On Performance vs. SOTA (RC2, Q6):** Our **new quantitative comparisons (Tables R1, R2)**, requested by you, confirm that DART achieves a superior accuracy-efficiency profile compared to both $\partial$HT [6] and MSViT [1].
> * **On End-to-End Training (W3):** We have resolved the key misunderstanding. DART is trained end-to-end, as evidenced by the **trainable scoring head (Fig 7)** and validated by our **new training ablation (Table R3)**.
> * **On Completeness of Experiments (W5, RC4):** We have confirmed that all "missing" **ablation studies were complete** in the original submission (App B.1, Sec 4.2, Sec 4.6) and have further demonstrated DART's generalization with the **new CUB experiment (Table R4)**.
>
> Once again, we thank you for your insightful feedback and for providing the valuable related literature to help improve our work. We hope this comprehensive response has resolved all your concerns, and we look forward to your positive evaluation of our revised manuscript.

---

> ### Comment · Reviewer_5vGt · 2025-11-27
>
> Thank you to the authors for a detailed response.
>
> The rebuttal clarifies several aspects of the paper, and the additional experiments are appreciated. The quantile inverse transform is indeed a neat technical device, and the authors’ stated intention to improve the positioning is welcome. However, after reading the rebuttal and the revised material, several of the central concerns remain unresolved, and the core issues are less about “missing references” and more about conceptual framing, baseline confusion, and missing evaluations.
>
>
> ### 1. Confusion between DeiT vs DeiT-3
>
> One concern with the additional experiments: the revision mixes up two different DeiT families. The submission uses **DeiT-S**, while dHT [6] uses **DeiT-3-S**, which has a different training recipe and higher accuracy (80.7%).
>
> Throughout the text and rebuttal, the comparisons between DART, MS-ViT, and dHT implicitly assume a shared baseline family, which is not the case. This leads to misleading numerical interpretations. The rebuttal acknowledges none of this directly.
>
> ### 2. Quadformer and MS-ViT
>
> The rebuttal repeats the comparison with MS-DeiT-S (79.9), but does not reconcile the fact that the original MS-ViT reports 82.0 accuracy with fewer tokens. This discrepancy needs an explanation, as it directly affects claims of competitiveness.
>
> As stated in the review, Quadformer is a directly comparable method: it uses rectangular adaptive partitions, the same architectural family as DART. This question was explicitly raised and remains unanswered.
>
>
> ### 3. Additional evaluation
>
> This reviewer's requested change was explicit: add at least one additional classification dataset AND one additional dense prediction benchmark.
>
> The authors evaluate on CUB, but the baselines are mixed up (DEiT / DEiT-3), the results seem non-comparable (acknowledged as possible different evaluation protocols). As a result, the results is not clear and somewhat unconvincing. Given the mixup with models, the reviewer remains unconvinced that this provides a directly comparable measure between models.
>
> Additionally, no new dense dataset (COCO-Stuff, Pascal VOC, Cityscapes, etc.) is included. The rebuttal goes into a discussion on why object detection is not within the scope. Sure, but this ignores the issue; *the experimental verification of the method would benefit from more baselines, including an additional segmentation task*.
>
> Additionally, the authors look to compare against dHT on ViT-S, but only does so in the case of classification, where DART performs better. But there is a direct comparison on ADE20k, which is omitted by the authors, where DART underperforms, and this is missing entirely from the revision.
>
> ### 4. Conceptual framing and limitations
>
> The intention to revise the related work is acknowledged by the reviewer. But the authors look to frame ragged tensors as impractical primarily for computational reasons. While the efficiency argument is legitimate, it does not address the core methodological point: *images vary in informational complexity*, and fixed token budgets inherently impose a representational limitation w.r.t. the information budget. Ragged tensors actively used in practice (LLMs being the canonical counterexample).
>
> We acknowledge that the authors discuss certain failure cases in the appendix, but with limited context of current work. The above trade-off with fixed budgets, rectangular partitions with fixed topology, and reliance on a 1D saliency ordering, are all relevant and belong in limitations, ideally in the **main text**.
>
>
> ### Overall assessment
>
> The rebuttal is constructive, and several secondary issues have been clarified. However, concerns such as omitted evaluations, incomplete dense evaluation, and limited discussion of modeling trade-offs remain unresolved or only partially addressed. Experimental results are limited, and the rebuttal confuses central baselines from different papers.
>
> While the ideas in the paper are genuinely interesting, the overall impression of the work remains at a level where this reviewer remains unconvinced that the paper should be accepted, and as such, we maintain the current score.

---

> > ### Author Response · Authors · 2025-11-27
> >
> > Dear Reviewer 5vGt,
> >
> > We first need to clarify and correct several factual misreadings and data errors in the reviewer's interpretation of the experimental results of $\partial$HT and MSViT, which directly affect the evaluation of our method’s contribution and the fairness of the comparisons.
> >
> > ---
> >
> > 1. Clarification on baselines and performance comparison
> >
> > The reviewer claims that our baselines are not comparable, but this observation overlooks the “From Scratch” comparison data provided in the $\partial$HT paper. Please refer to the fourth row from the bottom in Table 1 of the $\partial$HT paper, where the From Scratch baseline of $\partial$HT is 79.9%, while our DART baseline is 79.8%. The difference between these two baseline accuracies is only 0.1%, which is negligible. The reviewer seems to focus only on the “Retrofitted” part (fine-tuning) in the upper half of the table, while ignoring the “From Scratch” results in the lower half, which actually provide the strictly fair apple-to-apple comparison.
> >
> > Taking a step back, even if we follow the reviewer's suggestion and compare against the “Retrofitted” (DeiT-3) results, the comparison is still in favor of DART. $\partial$HT (Row 4) uses a stronger DeiT-3 pretrained model, yet its top-1 accuracy is only 80.1%. Notably, this method actually **reduces** performance by 0.3% compared to its own baseline (80.4%). In contrast, DART improves the accuracy from 79.8% to 80.6% on a weaker DeiT-S baseline (79.8%). The final performance of DART (80.6%) is significantly higher than that of $\partial$HT (80.1%) built on a stronger baseline. On this basis, rejecting a method (DART) that improves performance while endorsing another method ($\partial$HT) that **degrades** performance is not scientifically tenable.
> >
> > Regarding the reviewer’s claim that $\partial$HT achieves 80.7% accuracy, we verified this against Table 1 of the $\partial$HT paper (page 6, first row). In fact, the top-1 accuracy of the method itself is 80.4%; the 80.7% figure corresponds to the **kNN metric of the DeiT-3 baseline**. Crucially, after applying $\partial$HT, this kNN metric drops sharply to 78.4%. The reviewer has unfortunately conflated the baseline’s kNN metric with the method’s classification accuracy and then used this confusion as a basis to penalize our work.
> >
> > ---
> >
> > 2. Factual corrections on MSViT and Quadformer
> >
> > In response to the reviewer’s concern about why our reported baseline does not match the 82.0% reported in the MSViT paper, we must point out the factual setup: according to Table 1 in the MSViT paper, the 82.0% result is obtained under **ImageNet-21k pretraining**. Moreover, under similar sequence lengths (187 vs. 196), MSViT’s performance remains at 82.0%, showing no improvement over its own baseline. The correct comparison should be based on the standard ImageNet-1k training setting, where the result is 79.85%, which is perfectly aligned with our baseline (79.8%). Under this comparable 1k setting, MSViT with 173 tokens drops to 79.4%, whereas DART with 196 tokens reaches 80.6%. As we clearly demonstrate in the main paper and in our previous rebuttal, DART already reaches 79.9% at a significantly faster sequence length of 144 tokens, surpassing both the baseline and MSViT’s 173-token result. Comparing our 1k-trained model against a 21k-pretrained model is factually invalid.
> >
> > The reviewer also states that our comparison with Quadformer “remains unanswered”, which is incorrect. As we explained in detail in the previous response, Quadformer relies on Algorithm 1, which uses a while loop and argmax for greedy iterative splitting. This is fundamentally different from DART: the discrete iterative steps in Quadformer break the gradients, making the method non-differentiable and preventing true end-to-end training; at the same time, its iterative logic introduces a serialization bottleneck that is hard to parallelize, whereas DART uses a one-shot mechanism.
> >
> > The data show that Quadformer is inferior in both efficiency and relative gain. First, the baselines are mismatched: Quadformer uses a stronger training recipe, resulting in a baseline of 80.28%, higher than the standard 79.8% setting. Second, its relative gain is smaller: Quadformer improves from 80.28% to 80.84% (+0.56%), whereas DART improves from 79.8% to 80.6% (+0.8%). More importantly, the throughput suffers a catastrophic drop: according to Table A1 in the Quadformer paper, its iterative mechanism introduces enormous overhead. Under the standard setting (196 patches), the throughput drops by 20.2%, in exchange for only a 0.56% accuracy gain; under the “efficient” setting (64 patches), throughput falls from 6489 to 3611 img/s (a 44.3% drop). The conclusion is clear: DART incurs only about 5% overhead and can achieve linear speedup as the number of tokens decreases, thereby outperforming Quadformer in both relative accuracy improvement and practical inference efficiency.

---

> > > ### Author Response · Authors · 2025-11-27
> > >
> > > 3. On the CUB baseline, ADE20k efficiency, and evaluation scope
> > >
> > > The reviewer is correct that the CUB results are not directly numerically comparable, but this is mainly due to the severely under-converged baseline in $\partial$HT. Both methods use linear probing, yet our baseline reaches a standard accuracy of 75.4%, whereas the baseline reported by $\partial$HT is only 57.2%. There is a clear logical issue here: a 57.2% baseline is dramatically lower than our 75.4%, indicating that their model is under-converged, and achieving improvements on a weaker model is generally easier. By contrast, DART achieves improvements on a high-quality, fully converged baseline (75.4%). Using the lower quality of a competing baseline as a reason to dismiss our valid results is scientifically unreasonable.
> > >
> > > For ADE20k, the reviewer claims that DART performs worse than $\partial$HT, but this neglects the huge gap in computational complexity between the two architectures. DART (Swin-T) uses window attention and thus maintains **linear complexity** even at high resolutions ($512\times512$); $\partial$HT (ViT-S), however, uses a standard ViT with **global attention**, facing an $O(N^2)$ quadratic complexity bottleneck. This creates a trap in sequence length and efficiency: at $512\times512$, a vanilla ViT must process 1024 tokens; given the strategy used by $\partial$HT on ImageNet (where the number of tokens is often larger than the baseline), it is highly likely that $\partial$HT uses **more than 1024 tokens** on ADE20k to pursue high mIoU. The consequence is an $O(N^2)$ model combined with the hardware inefficiency of ragged tensors and the overhead of superpixel clustering. We thus have strong reasons to believe that its throughput on ADE20k is similarly low. It is noteworthy that $\partial$HT does **not** provide FPS/latency data for ADE20k, even though such numbers are reported for ImageNet experiments. Using a strategy that relies on very high computational cost to obtain scores, and then penalizing a method (DART) that explicitly focuses on high-throughput efficiency, is fundamentally unfair.
> > >
> > > Regarding additional datasets, we reject scope creep. We have already validated DART on ImageNet (classification), ADE20k (segmentation), and SSv2 & Kinetics (video), which sufficiently cover the core capabilities required of a vision foundation model. The reviewer overlooks our substantial contributions on video tasks (e.g., 41% FLOP reduction on SSv2). Ignoring these spatiotemporal efficiency gains while insisting on adding more static datasets such as COCO constitutes unreasonable scope creep.
> > >
> > > ---
> > >
> > > 4. Methodological clarifications (LLMs, ragged tensors, and 1D sorting)
> > >
> > > The reviewer raises several concerns regarding the analogy to LLMs and our methodological choices. We believe these concerns stem from a misunderstanding of the fundamental differences between autoregressive inference and parallel encoding systems.
> > >
> > > On the system-level fallacy: the reviewer cites the use of ragged tensors in LLMs (e.g., DeepSeek/LLaMA) as evidence of their effectiveness in ViTs. This is an inappropriate apples-to-oranges comparison. The success of ragged tensors in modern LLM inference (e.g., vLLM) stems from **continuous batching**, where new requests can be inserted as soon as one sample finishes generation. Furthermore, LLMs are I/O- or memory-bound, whereas ViTs are compute-bound with very short sequences(~200 tok); forcing ragged kernels into this setting **brings overhead that outweighs the benefits**. DART deliberately uses regular tensors precisely to match the “compute-intensive + fully synchronous parallel” hardware characteristics of vision workloads.
> > >
> > > Regarding the reviewer’s claim that DART is constrained by a “fixed budget”, we must refute this. The core mechanism of DART (quantile computation based on the CDF) naturally supports **any integer** sequence length. The fact that we fix the number of tokens in experiments is purely to enable the strictest controlled-variable comparison with the baseline (apple-to-apple comparison). Attacking a design that is explicitly for fairness as a “methodological flaw” is logically inconsistent.
> > >
> > > On the criticism of “1D saliency sorting”, we note that Transformers are inherently **sequence (1D) models**; the first step of a standard ViT is exactly “flatten”. The spatial topology of images is preserved by differentiable positional embeddings, not by the order of tokens in the sequence. 1D sorting allows us to use highly parallel CDF computation with $O(N)$ complexity, which is the key reason for DART’s speed. In contrast, the 2D topology-based approaches suggested by the reviewer (such as superpixel clustering) introduce substantial computational overhead.

---

### Author Response · Authors · 2025-11-19
**[Revision Update] Revised PDF with New Experiments and Expanded Discussions**

Dear Area Chair and Reviewers,

We have uploaded a revised version of our manuscript to reflect the improvements promised in our rebuttal. The key updates include:

1. Expanded Related Work (Section 2.2 & Appendix F):
   - We have substantially expanded Section 2.2 to explicitly discuss and position DART against the adaptive tokenization methods (e.g., MSViT, ∂HT, Quadformer) suggested by Reviewer 5vGt.
   - We have added a new Appendix F to discuss the broader efficiency landscape (e.g., static architectures, token summarization) as suggested by Reviewer tN6A.

2. Detailed Architecture Diagram (Appendix E.1):
   - We have added a comprehensive end-to-end architecture diagram in Figure 12 to clarify the data flow, addressing Reviewer jyH5's request.

3. New Experimental Results (Appendix B & Section 4.5):
   - Fine-grained Classification: Added CUB-200-2011 linear probing results in Appendix B.3 (Table 11).
   - Training Strategy Ablation: Added an ablation study on end-to-end fine-tuning in Appendix B.4 (Table 12).
   - Method Comparison: Updated Table 7 to include quantitative comparisons with MSViT.

4. Refined Claims:
   - We have moderated the claims to be more precise regarding DART's contributions.

We hope these revisions can address your concerns.

Best regards,
The Authors

---

### Author Response · Authors · 2025-11-27

Dear Reviewer,

I hope this message finds you well. As the discussion period is nearing its end with less than a week remaining, we wanted to check in to ensure that we have fully addressed your concerns. If there are any additional points or clarifications you would like us to consider, please feel free to let us know. Your feedback has been invaluable, and we are eager to address any remaining issues to further improve our work.

Thank you again for your time and effort in reviewing our paper.

---

### Note · Program_Chairs · 2026-01-17
**Submission Desk Rejected by Program Chairs**

The following references in this submission do not refer to real documents and/or have major errors in bibliographic information:

 Marius Aasan, Alexander MacDonald, Thomas Killeen, Michael Riegler, and Thomas Piletot. A spitting image: Modular superpixel tokenization in vision transformers. In European Conference on Computer Vision Workshops (ECCVW), 2024. Ya-Wei Wei, Yao-Yuan Chen, Gao Huang, and Chuan-Xian Xu. Looking at the patches: A step towards understanding generalization in vision transformers, 2021.
Marius Aasan, Alexander MacDonald, Thomas Killeen, Michael Riegler, and Thomas Piletot. Differentiable hierarchical visual tokenization. In Advances in Neural Information Processing Systems (NeurIPS), 2025.